# Behavioral changes during the COVID-19 pandemic decreased income diversity of urban encounters

Takahiro Yabe [1] ✉, Bernardo García Bulle Bueno[1], Xiaowen Dong[2,3], Alex Pentland [1,3] & Esteban Moro[1,3,4] ✉

Diversity of physical encounters in urban environments is known to spur economic productivity while also fostering social capital. However, mobility restrictions during the pandemic have forced people to reduce urban encounters, raising questions about the social implications of behavioral changes. In this paper, we study how individual income diversity of urban encounters changed during the pandemic, using a large-scale, privacy-enhanced mobility dataset of more than one million anonymized mobile phone users in Boston, Dallas, Los Angeles, and Seattle, across three years spanning before and during the pandemic. We find that the diversity of urban encounters has substantially decreased (by 15% to 30%) during the pandemic and has persisted through late 2021, even though aggregated mobility metrics have recovered to pre-pandemic levels. Counterfactual analyses show that behavioral changes including lower willingness to explore new places further decreased the diversity of encounters in the long term. Our findings provide implications for managing the trade-off between the stringency of COVID-19 policies and the diversity of urban encounters as we move beyond the pandemic.

Cities are the central drivers of economic productivity and innovation owing to their capacity to foster dense social connections through physical encounters[1–3]. Among the various characteristics of social connections and network structures, empirical studies have shown that the diversity of networks is a significant predictor of economic growth and recovery[4,5]. Moreover, integrated community networks and the inherent social capital are crucial for resilience to shocks such as natural hazards[6,7]. The lack of community support could lead to inequitable access to urban amenities and services, ultimately affecting social, economic, and health outcomes of people living in urban areas[8]. However, in addition to rising inequality and segregation[9], the COVID-19 pandemic and the consequential countermeasures including mobility restrictions have posed significant challenges for maintaining both the quantity and quality of such physical encounters in cities.

Large-scale location data (e.g., CDRs[10,11], credit card data[12], and social media[13]) have been used to understand the nature of physical encounters of people in cities[14,15]. Recently, such mobility datasets have been used to measure and understand the diversity of encounters in cities[16–18], by measuring the homophily of co-locations at points-of-interest (POIs) during daily routines. A study using mobile phone data in 10 American cities revealed that peoples' mobility behavior, as opposed to their residential locations, account for 55% of urban segregation (which is an inverse metric of diversity)[19]. Another study using Foursquare data revealed that people mostly visit places in their own socioeconomic status, occasionally visiting venues from higher income places[20]. Compared to analysis limited to quantifying static residential segregation measures using census data[21], such studies based on mobility data have

[1]Institute for Data, Systems, and Society, Massachusetts Institute of Technology, Cambridge, MA 02139, USA. [2]Department of Engineering Science, University of Oxford, Oxford OX2 6ED, UK. [3]Media Lab, Massachusetts Institute of Technology, Cambridge, MA 02139, USA. [4]Grupo Interdisciplinar de Sistemas Complejos (GISC), Departamento de Matemáticas, Universidad Carlos III de Madrid, 28911 Leganés, Madrid, Spain. ✉e-mail: tyabe@mit.edu; emoro@mit.edu

provided a more comprehensive understanding of income segregation in urban environments.

Various non-pharmaceutical interventions imposed by governments and agencies have enforced people to substantially change their lifestyles and reduce daily activity patterns, reducing trips to urban amenities such as restaurants, bars, and entertainment establishments[22]. While such behavior changes have had significant impacts on our physical health and activities[23,24] and mental wellbeing[25], studies have also suggested their impact on social encounters in urban environments, affecting the social fabric of cities we live in[26]. The aftermath of the pandemic has brought also significant changes in behavior in our cities, including less use of public transportation[27], more hours working from home[28], and higher usage of online food and goods delivery services[29]. There have been many studies in the fields of civil engineering, transportation engineering, geography, and urban planning that have analyzed the short-term mobility dynamics during the early stages of the pandemic and lockdown[30–35], socioeconomic disparities in response to lockdowns[36–38], and its impact on disease spread using location data collected via mobile phones[39,40]. Moreover, several studies have investigated the longitudinal changes in transportation and mobility patterns in urban areas[41].

However, little is understood about how much longitudinal effects the pandemic has had on the quantity and quality of our encounters in urban environments. Measuring the dynamics and potential causes of fluctuations in the diversity of urban encounters across different periods of the pandemic could be valuable in understanding the long-term impacts of the pandemic on cities, and for developing resilient policies to better prepare for future outbreaks. To this end, this study contributes to bridging this knowledge gap by analyzing how the income diversity of urban encounters has changed before and during the pandemic across a 3 year timespan, using a large-scale, privacy-enhanced mobility dataset of more than one million anonymized mobile phone users in four large US cities. We find that the diversity of urban encounters has substantially decreased during the pandemic and has persisted through late 2021, despite the recovery of aggregated mobility metrics. Furthermore, we conduct counterfactual analyses to show that behavioral changes including lower willingness to explore new places and changes in visitation preferences substantially decreased the long-term diversity of encounters. Our findings suggest that the pandemic could have long-lasting negative effects on experienced income diversity, and highlight the importance of urban policies to recover the diversity of urban encounters as we move beyond the pandemic.

## Results

Using a large and longitudinal dataset of GPS location records in four major metropolitan areas in the US across more than three years, we analyze how experienced income diversity of urban encounters has changed during different periods of the COVID-19 pandemic. Specifically, we analyze the dynamics of income diversity of encounters at the level of individual places (points-of-interest; POIs) and individual users in cities. We seek to identify behavioral changes that were the cause of such long-term changes, and we further unravel the socio-demographic, economic, and behavioral characteristics that explain the spatial heterogeneity in decreased diversity. Mobility data was provided by Spectus, who supplied anonymized, privacy-enhanced, and high-resolution mobile location pings for more than 1 million devices across four U.S. census core-based statistical areas (CBSAs) (Supplementary Table S2). All devices within the study opted-in to anonymized data collection for research purposes under a GDPR and CCPA compliant framework. Our second data source is a collection of 433K verified places across four CBSAs, obtained via the Foursquare API. The robustness of the results on experienced income diversity against the choice of places dataset was checked using the

ReferenceUSA Business Historical Data[42] (Supplementary Note 1). Post-stratification techniques were implemented to ensure the representativeness of the data across regions and income levels (Supplementary Note 2).

To analyze the income diversity of urban encounters, each anonymized individual user in the dataset was assigned a socio-economic status (SES) proxy, estimated from their home census block group (CBG) using the 2016-2020 5-year American Community Survey (ACS) (Supplementary Note 1). The approximate home area of each individual user was estimated by Spectus at the granularity of CBGs using their most common location during the nighttime, between 10 p.m. and 6 a.m. every week. Individuals were then categorized into four equally sized SES quantiles according to the median household income of their home CBG. The results on decreased experienced income diversity were robust against the number of income quantile categories used (Supplementary Note 1). Only users who were observed more than 300 minutes each day were used for the analysis to remove users with substantial missing data. Stays (stops) longer than 10 minutes and shorter than 4 hours were then extracted from the dataset, and each stay was spatially matched with the closest place locations within 100 meters to infer stays at specific POIs. The results on experienced income diversity were robust against the choice of data filtering parameters and spatial threshold parameters for visit attribution (Supplementary Note 1).

Given the estimated SES quantiles of individual users and the visited POIs, we measured the experienced income diversity at each place $\alpha$ (denoted as $D_\alpha$) and experienced by each individual $i$ (denoted as $D_i$). $D_\alpha$ measures the evenness of the time spent by people from different income quantiles at each place, and $D_i$ measures the evenness of time spent with people from different income quantiles for each individual (see Methods and Supplementary Notes 3.1 and 3.2). For places, $D_\alpha = 1$ when the place is fully diverse, with 25% of time spent by people from each of the four income quantiles, and $D_\alpha = 0$ when the place is visited by members of only a single income quantile. Similarly, to calculate the diversity of individuals $D_i$, we measure the exposure of the individual $i$ to each income quantile $q$ across all the places $\alpha$ the individual has visited. The robustness of the results to the choice of diversity metric was tested (Supplementary Note 3). The diversity measures were computed for each 2-month moving window to ensure a sufficient number of visits to POIs, and were deseasonalized using monthly trends observed in 2019. The panels in Fig. 1a show how experienced income diversity at places around the Boston and Cambridge area substantially decreased during the first wave of the pandemic. The diversity of encounters gradually recovers, however, not fully even after more than 1 and a half years from the lockdown, in October 2021. Similar patterns can be observed in all three other cities in the study. The maps highlight the significant spatial heterogeneity of experienced income diversity (e.g., Back Bay area is more diverse compared to the suburban areas), which is further investigated in the later sections.

### Diversity of urban encounters has decreased during the pandemic

The pandemic substantially changed people's mobility patterns in the early waves, as documented in previous studies using mobility data (e.g.,[43]). However, several individual mobility metrics indicate that individual-based mobility patterns have returned back to pre-pandemic levels by late 2021. Figure 1b shows monthly average values of several individual mobility metrics across the two years of 2020 and 2021. Mobility metrics, more specifically the daily number of visits per individual, daily amount of time spent at POIs per individual, average dwell time spent per visit, and number of visited unique POIs per individual, have all returned back to pre-pandemic levels (annotated by horizontal dashed lines) by late 2021 in all four CBSAs. The drop in the rate of visits to POIs as well as the duration of visits to POIs

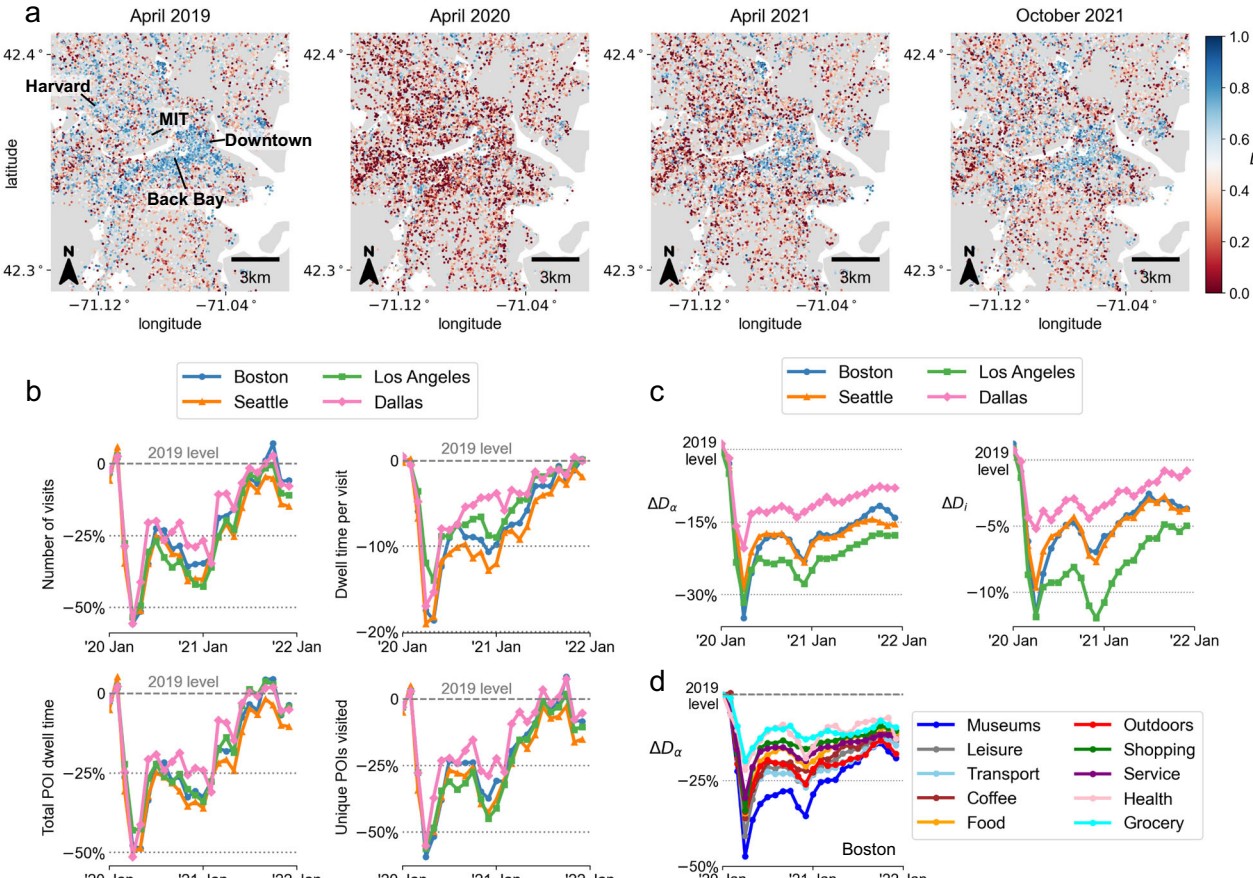

**Fig. 1 | Diversity of urban encounters has decreased during COVID-19. a** Map shows that the income diversity of encounters in places in the Boston and Cambridge area decreased during the pandemic. Diversity gradually recovers with reopening, albeit not fully compared to pre-pandemic levels, even in October 2021. **b** Aggregate mobility metrics, such as the daily number of visits per individual, daily amount of time spent at POIs, and number of visited unique POIs have all returned back to pre-pandemic levels (i.e., 2019 average values) by late 2021. **c** Despite the recovery in mobility statistics, the diversity of encounters experienced at places and by individuals has decreased and have not recovered back to pre-pandemic levels. The differences are statistically significant due to the large number of POIs (50–200 K depending on the city) and users (140–450K). On average, the standard error of the average individual and place segregation metrics are 0.13% and 0.14% of the actual values, respectively, and are too small for visibility. **d** Experienced income diversity decreased in all major place categories both in the short-term (e.g., April 2020) and long-term (e.g., October 2021) in all cities. Grocery stores consistently experienced the least effects of the pandemic while museums, leisure, transport, and coffee places saw the largest decrease. Maps were produced in Python using the TIGER shapefiles from the U.S. Census Bureau[61].

during the earlier stages of the pandemic agree with the findings in previous studies[44], however, our analysis extends the analysis to two years into the pandemic and confirms how activity patterns have recovered back to pre-pandemic levels by October 2021. The mobility data confirms that people have resumed spending time outside their homes and visiting different POIs, similar to before the pandemic.

Given the recovery of aggregate mobility metrics, one could expect the income diversity of encounters to also return back to pre-pandemic levels by late 2021. However, as shown in Fig. 1c, the income diversity experienced at places and by individuals is consistently lower than the pre-pandemic levels for all four cities even after 2 years into the pandemic. Absolute values of $D_\alpha$ and $D_i$ are shown in Supplementary Fig. S15. Cities experience the most decrease in diversity in April 2020, 30% lower than pre-pandemic levels during the lockdown. A second peak in the loss of diversity is observed in late 2020, which corresponds to the increase in cases due to the first SARS-CoV-2 variant. Despite the recovery of individual mobility metrics as shown in Fig. 1b, income diversity of encounters is still around 10% less than pre-pandemic levels even by late 2021. $D_i$ is a convoluted version of the $D_\alpha$ for all places $\alpha$ visited by $i$, which considers the ratio of the sum of stay duration across all income quantiles. Thus, although each place $\alpha$ could significantly lose

income diversity during the pandemic due to a decrease in visits, individuals are likely to visit multiple places in a given month, which results in a higher $D_i$ than $D_\alpha$. The decrease in experienced income diversity was robust to the choice of diversity metrics, such as the entropy of income quantiles for encounters at places and for individuals (Supplementary Note 3.3).

Dissecting the place-based diversity results into POI categories, we further observe that diversity in places in Boston decreased in all POI categories both in the short-term (e.g., April 2020) and long-term (e.g., October 2021) in Fig. 1d. Especially, 'Museums', 'Leisure', 'Transportation', and 'Coffee' places had the largest decrease in diversity, while 'Grocery' places consistently experienced the least effects of the pandemic. This is consistent with the fact that we observe the number of visits to follow similar patterns, where we see a decrease during the early stages of the pandemic and gradual recovery by late 2021 for all POI categories, except grocery stores, which experienced no reduction in the number of visits even during the first waves. This suggests that the reduction in the number of visits indeed is one of the factors that cause the decrease in the diversity of encounters. In the following section, we employ a counterfactual analysis approach to further understand why the diversity of encounters has consistently decreased during the pandemic.

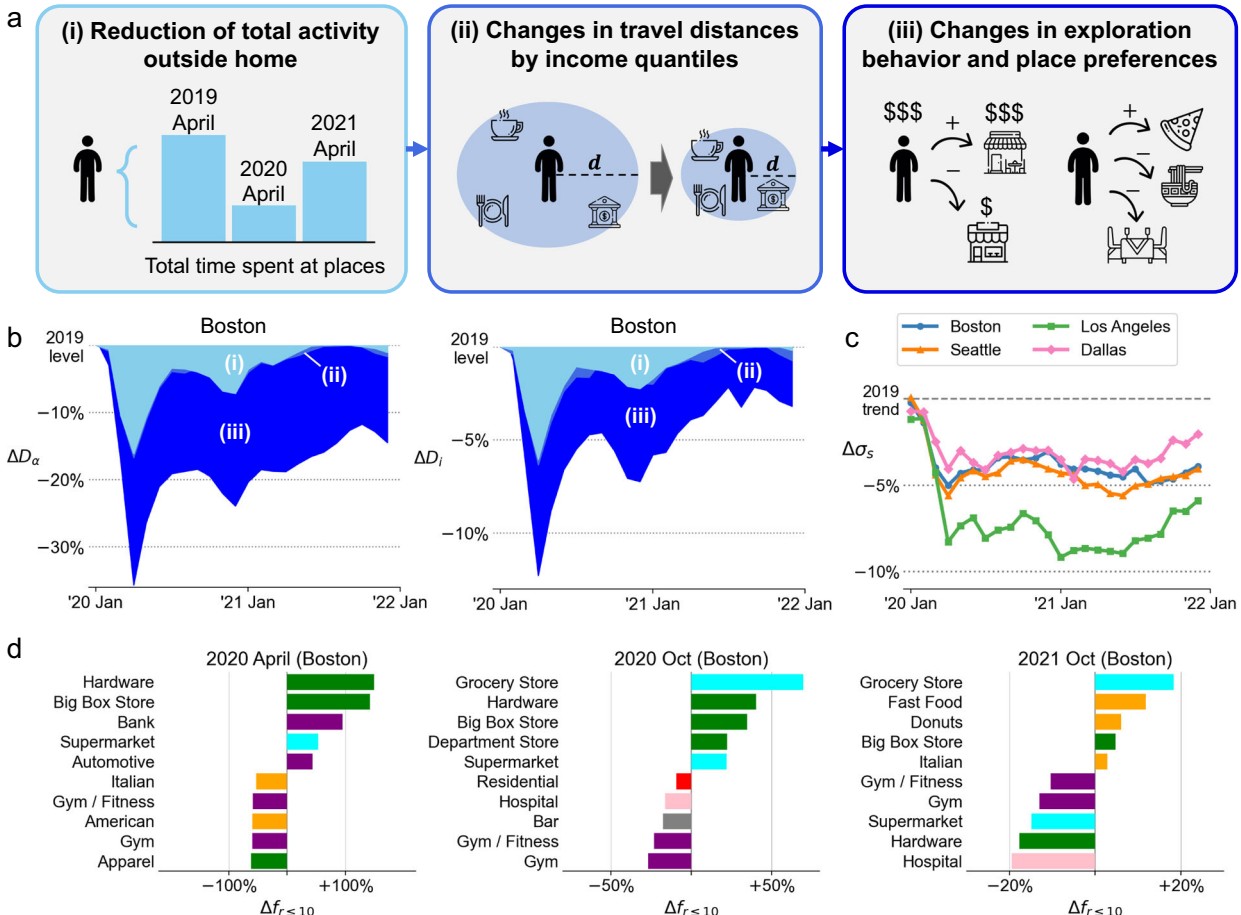

**Fig. 2 | Behavioral changes decreased experienced income diversity in cities.**
**a** Three hierarchical levels of behavioral changes were simulated to understand why experienced income diversity decreased: (i) reduction in total outside activity by income groups, (ii) changes in traveled distances by income groups, and (iii) microscopic changes in mobility behavior, including exploration behavior and place sub-category preferences. **b** Decrease in the diversity of encounters for places and individuals decomposed into the three behavioral factors for Boston. Counterfactual simulations show that reduction in total activities (i) in the short-term, and changes in exploration and place preferences (iii) in the long-term, were the major factors that decreased diversity. **c** Social exploration, which quantifies the probability of visiting a new place where the individual is a minority in terms of income groups, decreased during the pandemic compared to 2019 trends in all four cities. **d** POI subcategories that were more (and less) visited in different periods during the pandemic. Colors correspond to the major POI categories used in Fig. 1d. More routine locations such as grocery stores, big box stores, and fast food places are visited more, while places such as gyms and fitness places are visited less during the pandemic. Figure (**a**) was designed using icons from Flaticon.com created by Freepik, Eukalyp, and kerismaker. Maps were produced in Python using the TIGER shapefiles from the U.S. Census Bureau[61].

## Behavioral changes decreased experienced income diversity in cities

To investigate the behavioral factors that led to the consistent decrease in income diversity experienced at places and by individuals, we consider three possible hierarchical levels of changes in the behavior of individuals due to the pandemic. As illustrated in 2A, the pandemic led, especially during the beginning of the pandemic, to a (i) reduction in the total amount of time spent at places outside homes and workplaces. Moreover, due to stay-at-home orders and also reluctance towards long-distance trips compared to before, we also consider (ii) changes in travel distances for each income quantile. Since some types of activity categories were particularly affected by social-distancing policies, we also consider changes in visits to major activity categories and traveled distances for each income quantile, shown in Supplementary Note 4. Finally, we also consider the possibility of (iii) microscopic changes in place preferences, including changes in exploration behavior and visitation patterns across place subcategories.

To disentangle the relative impacts of these behavioral changes, we created different counterfactual mobility datasets. For example, to estimate the effects of the reduction of total activity time on the loss of

diversity, we randomly removed visits from pre-pandemic data (e.g., 2019 April) to create a counterfactual mobility dataset that contains the same total visit duration at places during the pandemic (e.g., 2020 April) (see Methods and Supplementary Note 4). The resulting generated counterfactual data can answer the question of 'how would the income diversity change if people just simply decreased the number of visits to places from pre-pandemic (2019) levels?'. By comparing the place and individual-based diversity measures computed from the actual and the counterfactual mobility datasets, we are able to delineate the effects of activity reduction on the decrease in diversity. Similarly to measure the effects of (ii) changes in traveled distances by income quantiles, we extended the previous counterfactual to have the same total visit duration by distance ranges for each income quantile (see Methods and Supplementary Note 4). Simulations were run 10 times each to ensure robust results.

Figure 2b shows the decreased diversity experienced at places and by individuals decomposed into the three behavioral factors (full results shown in Supplementary Fig. S23). The counterfactual simulations show that (i) reduction in total activities caused around 50% of the decrease in diversity during the first pandemic wave, however, decreases to almost 2% by late 2021 when mobility metrics have

recovered back to normal, as shown in Fig. 1b. Although we observe different rates of dwell time decrease and recovery across income quantiles, where the higher income populations disproportionately reduce dwell times at places than lower income populations, the overall diversity measures are not affected since the relative mixing of population groups across income groups is consistent (Supplementary Note 4.2). Changes in distance distributions, where people prefer trips to closer places during the pandemic, have slight negative effects on the income diversity of encounters. Surprisingly, changes in dwell time duration at major activity categories had no effects on the experienced income diversity metrics (Supplementary Note 4.2).

Heterogeneity in activity reduction rates across income quantiles and changes in traveled distances explain around 55% of the decreased diversity during the first wave of the pandemic, however, the remaining 45% is due to more microscopic, place-based preference changes. These effects become the single dominant factor in the later stages of the pandemic. To identify the changes in the mobility behavior during the pandemic, we fit the social exploration and preferential return (Social-EPR) model[19,45] to the data for each period and assess the model parameters (see Supplementary Note 4.3). Among the parameters of the social-EPR model, the parameter which changed the most between before and during the pandemic was the social exploration parameter $\sigma_s$, as shown in Fig. 2c and Supplementary Fig. S26. Social exploration $\sigma_s$ measures the probability of an individual visiting a place where their income group is not the majority income quantile group when they decide to explore a new place. During the pandemic, people's willingness to socially explore substantially decreased compared to the 2019 levels (horizontal dashed line) in all four cities, leading to less experienced diversity.

Furthermore, we observe changes in place level preferences across POI subcategories. Sub-category popularity $f_r$ is measured by computing the probability that a POI sub-category is included in an individual's top $r$ most frequently visited places. Figure 2d and Supplementary Fig. S27 show the POI subcategories which were more (and less) visited in different periods during the pandemic compared to 2019 levels. Hardware stores, big box stores, and grocery stores (in October 2020 and 2021) were POI subcategories that gained popularity during the pandemic, and gyms, movie theaters, and American food places were subcategories that were less visited frequently. Taken together with the results that controlling by major activity categories did not explain additional decreased diversity to scenario (ii) as shown in Supplementary Note 4.2, this result shows that people have not changed their proportion of time spent for major activity categories, but have changed which specific types of places they visit within each major activity (e.g., less time at American restaurants, but more time at fast food and donut stores). To summarize, not only a reduction in activity, but also microscopic behavioral changes especially during the later stages of the pandemic, including less exploration and shift in preferences, led to decreased diversity in urban encounters.

## Spatial and socioeconomic heterogeneity in decreased diversity

Which sociodemographic groups and areas were more affected by the decrease in experienced income diversity? To understand the heterogeneity in decreased diversity, the mean CBG-level experienced income diversity of all individuals living in the CBG were computed for each CBG in the four CBSAs, thus $D_{CBG} = \frac{1}{|N_{CBG}|}\sum_{i \in N_{CBG}} D_i$, where $N_{CBG}$ denotes the set of individuals living in the corresponding CBG. By visualizing $\Delta D_{CBG} = 100\% \times (D_{CBG} - D_{CBG}^{2019})/D_{CBG}^{2019}$ in the Boston-Cambridge-Newton CBSA in Fig. 3a (and other CBSAs in Supplementary Fig. S28), we observe spatial heterogeneity in the changes in diversity in the early stages of the pandemic, however, more homogeneity in the long term. The insets also show the magnitude of $\Delta D_{CBG}$ decreasing as cities recover from the pandemic. The correlation

between $D_{CBG}$ in April 2020 and $D_{CBG}$ in April 2019 is much smaller ($R^2 = 0.37$) than for October 2021 and October 2019 ($R^2 = 0.71$), indicating a larger heterogeneity in $\Delta D_{CBG}$ during the earlier stages of the pandemic (Supplementary Fig. S29).

To understand the spatial and sociodemographic heterogeneity in the decreased diversity of encounters during the pandemic compared to 2019, we model $D_{CBG}$ and its difference $\Delta D_{CBG}$, using a simple regression model (see Methods and Supplementary Note 5). We include variables describing the places visited by the residents in the CBG (in 2019), mobility metrics including the average total traveled distance and radius of gyration (in 2019), and sociodemographic and economic characteristics of the CBG, including its population density, median income, age and race composition, and transportation behavior (e.g., public transportation usage), all of which were standardized (Supplementary Table S3). Regression analysis was conducted for each month, including all four cities. To control for the difference between areas across and within the metropolitan areas, we include geographical fixed effects at the level of Public Use Microdata Areas (PUMAs), which typically span around 20km and contain a residential population of 150 thousand people. Detailed summary statistics, collinearity and correlations between variables, variance inflation factor analysis, and full regression results can be found in Supplementary Note 5.

Figure 3b shows the adjusted $R^2$ of regression models for $D_{CBG}$ and $\Delta D_{CBG}$, respectively, across different periods. The three groups of variables (places visited, geographical mobility, and residence and demographics) explain around 60% to 70% of the variance of experienced income diversity ($D_{CBG}$), which agrees with previous findings[19] (Supplementary Tables S4–S6). However, the difference in diversity from 2019 levels ($\Delta D_{CBG}$) has lower explained variance (at most $R^2 = 0.31$), and also decreases where there is no pandemic outbreak. In the long-term (October 2021), the regression model has low explained variance ($R^2 = 0.11$), indicating that regions homogeneously became less diverse, irrespective of sociodemographic or behavioral characteristics of the areas. Figure 3c shows the factors that were most important in explaining the variance of $\Delta D_{CBG}$ in the months where $R^2$ was relatively high (April, May, December 2020 and January 2021) (Supplementary Tables S7–S10). The highlighted regression coefficients suggest that whenever there is an outbreak, areas with a higher population density and higher proportion of working-age populations (age 25–64), higher reliance on public transport, and larger movement range (radius of gyration) experience the largest decrease in income diversity of encounters.

## Trade-off between income diversity of encounters and stringency of policy measures

From a public policy perspective, an important and interesting question is to understand how COVID-19 containment measures, including lockdowns, school and workplace closures, and restrictions on public gatherings, have affected resulted in the loss of diversity in urban encounters. To measure the relationship between the stringency of COVID-19 measures and experienced income diversity, we utilize the COVID-19 Stringency Index[46] (Supplementary Fig. S37), which is a composite measure of nine response metrics, including school and workplace closures, restrictions and cancellation of public events and gatherings, and restrictions on movement and travel (See Supplementary Note 6).

Figure 4 shows the relationship between the stringency of COVID-19 policies and the decrease in the diversity of urban encounters. In all four cities we observe statistically significant ($p < 0.01$) and strong negative correlation ($\rho(SI_{CBSA}, \Delta D_{CBSA}) \in [-0.9, -0.73]$). The robust negative correlations suggest a strong trade-off relationship between experienced income diversity and COVID-19 policy and outbreak intensity in all cities. The decrease in diversity become pronounced during COVID-19 outbreaks, especially during the first pandemic wave

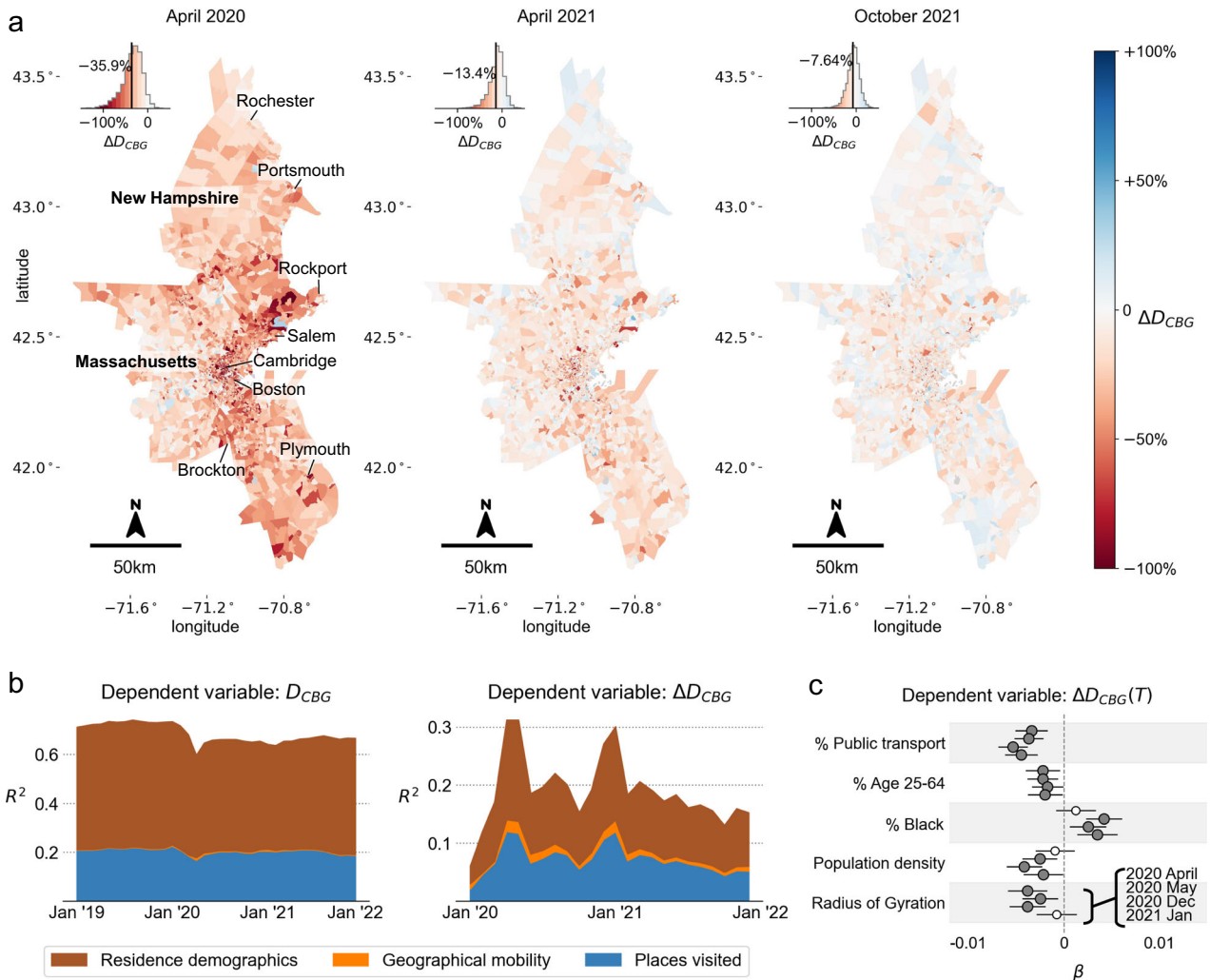

**Fig. 3 | Spatial and socioeconomic heterogeneity in decreased diversity. a** Maps show changes in mean experienced income diversity on the census block group (CBG) levels in the Boston CBSA for three different time periods (April 2020, April 2021, and October 2021), compared with the corresponding months in 2019. Insets show histograms of differences in experienced income diversity $\Delta D_{CBG}$. **b** Adjusted $R^2$ of regression models for $D_{CBG}$ and $\Delta D_{CBG}$, respectively, across different time periods. The three groups of variables (places visited, geographical mobility, and residence and demographics) explain around 55% to 70% of the variance in experienced income diversity. However, the same variables explain a much lower variance of $\Delta D_{CBG}$, indicating that regions became less diverse homogeneously.

**c** Regression coefficients that explain the heterogeneity in $\Delta D_{CBG}$ for the four different time periods where the $R^2$ was relatively higher. Filled variables are statistically significant at the $P < 0.05$ threshold. CBGs which have a higher proportion of public transport use, higher population density, and a larger proportion of the working population (age 25–64) had a larger decrease in experienced income diversity ($n = 427,776$ census block groups across 4 cities). The statistical tests were two-sided and data are presented as mean values and 95% confidence intervals. Maps were produced in Python using the TIGER shapefiles from the U.S. Census Bureau[61].

(red plots) in Boston and Seattle, and during the second pandemic wave (orange plots) in Los Angeles, where the number of cases and deaths were substantial in the respective cities. Moreover, for Boston, Seattle, and Los Angeles, even though the Stringency Index has decreased to around 20 in late 2021 (which indicates already less strict policies in place), the decrease in experienced income diversity is positive, suggesting that the COVID-19 pandemic may have had a long-lasting decreasing effect on the income diversity of urban encounters. Regression results using additional exogenous variables such as the number of COVID-19 cases and deaths on the federal and local (CBSA) levels are shown in Supplementary Note 6.1 and Supplementary Fig. S38. Since $\Delta D_{CBSA}(t)$ is temporal data with autocorrelation, we tested ARIMA type models as well. For Boston and Seattle, the moving average component was significant, however, the estimated coefficients of the stringency index were found to be robust. The temporal effects for Los Angeles and Dallas were insignificant (see Supplementary Note 6.2).

## Discussion

Cities around the world currently face a wide array of challenges, ranging from combating inequality in wealth and economic opportunities[47], to avoiding catastrophic outcomes caused by climate change-induced disasters[48]. Improving the inherent social capital of local communities and neighborhood networks, which are the fundamental units of collective decision making and support, is crucial for tackling these complex and global-scale societal challenges. With many cities expanding and urban inhabitants increasing at an unprecedented pace, the importance of promoting diverse encounters has never been higher[49]. Previous literature shows that physical co-location and encounters are known to be significant factors[50] and predictors[51] for real world friendship formation, accounting for around 30% of new friendship additions[52]. Moreover, the diversity of our social encounters has been shown to help communities to acquire social capital and resilience to natural hazards[6], and to foster economic growth[53]. Therefore, a decrease in experienced income diversity over

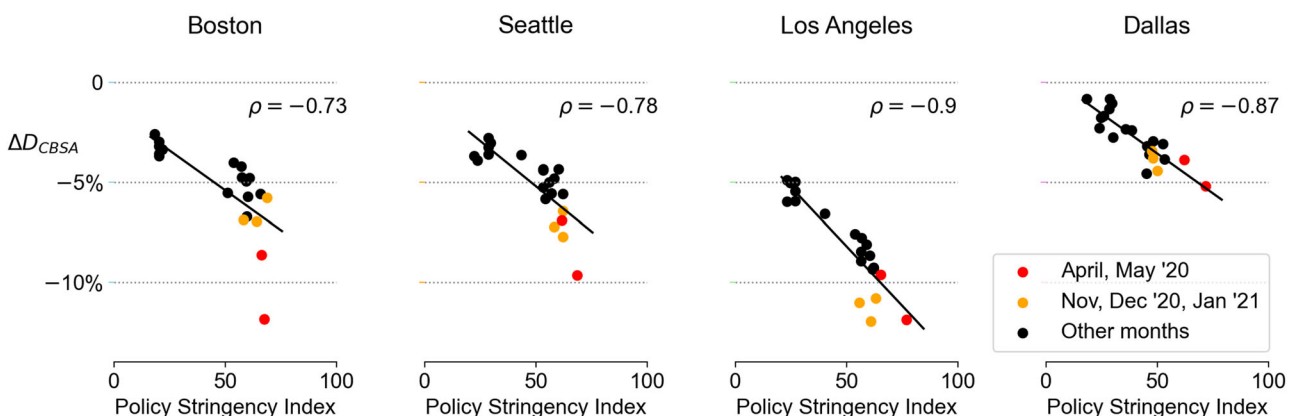

**Fig. 4 | Trade-off between decreased income diversity of encounters and stringency of COVID-19 policies.** Decrease in income diversity of encounters $\Delta D_{CBSA}$ has a strong and significant correlation $\rho$ with the stringency of COVID-19 measures (which is a composite measure of nine response metrics, including school and workplace closures, restrictions, and cancellation of public events and gatherings, and restrictions on movement and travel[46]) in all four CBSAs, with outliers during the pandemic waves, especially in Boston, Seattle (first wave; in red) and Los Angeles (second wave; in orange).

the long term could have substantial cumulative effects on the number and diversity of friendship ties, leading to more income segregation and polarization.

In this study, we make three important contributions towards understanding the dynamics of experienced income diversity during and beyond the COVID-19 pandemic. First, we empirically revealed that physical encounters in US cities have indeed become less diverse than pre-pandemic levels even two years after the first case in the US, despite an almost full recovery in aggregate mobility statics (e.g., number of visits per day). Second, we identified key behavioral changes that resulted in lower income diversity of encounters during the pandemic, including the consistent decrease in the exploration of socially diverse places and shifts in visitation preferences. Third, comparative analysis with COVID-19 policies suggested a strong trade-off relationship between COVID-19 policy stringency and experienced income diversity. Thus, although social-distancing policies helped to mitigate the propagation of the epidemic, they also had negative effects on the social fabric of our cities. These insights, which are extremely difficult to quantify using traditional residence-based measures, collectively allow us to understand how and why urban encounters have become less diverse due to the pandemic.

Studies have suggested that while the development of effective vaccines has successfully suppressed the mortality rates of COVID-19, the new behavioral habits and social norms that we have acquired during the pandemic, such as higher rates of working from home, and dramatic changes in physical activity, sleep, time use, and mental health[54], could have a long-lasting impact on society[28]. Behavioral changes that were observed in this study, such as less social exploration when visiting new places and changes in place preferences, may also remain for a long period due to persistent fear of infections. Our results suggest that policy interventions on urban mobility that enable populations of different income groups to increase physical encounters should also target and evaluate the recovery of social exploration to potentially improve experienced income diversity after the pandemic. Other studies have recently highlighted the importance of these weak ties on career success[55] and economic mobility[56]. Such public interventions include the introduction of fare-free transit systems and the development of public spaces that aim to lower the cost of transport and opportunities from low-income areas to high-income areas.

The results of our study should be interpreted in light of its limitations. Although we have tested the robustness of the used mobile phone data from various aspects and applied a post-stratification strategy to overcome sampling biases, there may be overlooked biases arising from uncertainty in data collection algorithms, such as the

frequency and timing of location data collection[57,58]. Also, another limitation of the mobility data is that we are not able to identify the purpose of visits or the types of the encounters, for example, whether it is a co-visitation at a cafe where no conversations take place or a cocktail party where strangers meet and have a conversation over a common topic. Therefore, the metrics computed in our study should be interpreted as a proxy for all meaningful encounters, and as a bound for experienced income diversity. Regarding the study design, we focus on experienced income diversity and not other socio-economic and demographic dimensions, including racial diversity[13,59]. The methods and approaches may be applied to other socio-demographic data available in the American Community Survey to understand the dynamics of these other types of social diversity experienced in cities.

## Methods
### Mobility data
We utilize an anonymized location dataset of mobile phones and smartphone devices provided by Spectus Inc., a location data intelligence company that collects anonymous, privacy-compliant location data of mobile devices using their software development kit (SDK) technology in mobile applications and ironclad privacy framework. Spectus processes data collected from mobile devices whose owners have actively opted in to share their location, and require all application partners to disclose their relationship with Spectus, directly or by category, in the privacy policy. With this commitment to privacy, the dataset contains location data for roughly 15 million daily active users in the United States. Through Spectus' Social Impact program, Spectus provides mobility insights for academic research and humanitarian initiatives. All data analyzed in this study are aggregated to preserve privacy. The home locations of individual users are estimated at the CBG level using different variables including the number of days spent in a given location in the last month, the daily average number of hours spent in that location, and the time of the day spent in the location during nighttime. See Supplementary Note 1.1 for more details. The representativeness of this data has been tested and corrected in Supplementary Note 2 using post-stratification techniques. Since the data used was anonymized and spatially aggregated at places, categories, or census areas, we were granted an Exemption by the MIT Committee on the Use of Humans as Experimental Subjects (COUHES protocol #1812635935) and its extension #E-2962.

### Estimation of stays at places
Stops, which are location clusters where individual users stay for a given duration, are estimated using the Sequence Oriented Clustering

approach[60]. The stops are attributed to places (or points-of interest; POIs) by simply searching for the closest place from the stops within a 100 m radius. The robustness of the estimated experienced income diversity to this spatial parameter was tested in Supplementary Note 1.2. Moreover, the robustness of the results on experienced income diversity against the choice of place datasets was tested using the ReferenceUSA dataset[42] in Supplementary Note 1.3. Stays between 10 minutes and 4 h, of individuals who were observed for more than 300 min each day were used for the analysis. The results were shown to be robust against the choice of these temporal parameters in Supplementary Note 1.5.

### Income diversity of encounters

To measure the income diversity of encounters experienced at each place $\alpha$ in each city, we compute the proportion of total time spent at place $\alpha$ by each income quantile $q$, $\tau_{q\alpha}$. Income thresholds for the quantiles are chosen based on the income distributions in each city. We checked that the results on experienced income diversity are independent of the choice of the number of income quantiles in Supplementary Note 1.4. We define full diversity of encounters at a place when people from all income quantiles spend the same amount of time, $\tau_{q\alpha} = \frac{1}{4}$ for all $q$. Using the metric used to compute income segregation in urban encounters in previous studies[19], we define the income diversity experienced at each place $\alpha$, $D_\alpha$ as a measure of evenness of time spend by different income quantiles $D_\alpha = 1 - \frac{2}{3}\sum_q |\tau_{q\alpha} - \frac{1}{4}|$. The diversity measure is bounded between 0 and 1, where $D_\alpha = 0$ means there is no diversity (the place is visited by people from only one income quantile), and $D_\alpha = 1$ indicates that all income quantiles spent an equal amount of time at the place. Similarly for individuals, given the proportion of time individual $i$ spent at place $\alpha$, $\tau_{i\alpha}$, the individual's relative exposure to income quantile $q$, $\tau_{iq}$ can be computed by $\tau_{iq} = \sum_\alpha \tau_{i\alpha}\tau_{q\alpha}$. Then, the income diversity experienced by individual $i$ can be measured using the same equation used for places $D_i = 1 - \frac{2}{3}\sum_q |\tau_{iq} - \frac{1}{4}|$. Most of the results in the main manuscript are shown by percentage differences, which are computed by $\Delta D_i(t) = \frac{D_i(t) - D_i(2019)}{D_i(2019)} \times 100(\%)$, where $D_i(2019)$ is the mean income diversity of encounters across all individuals observed on the same month as $t$ in 2019, before the pandemic. Results in Supplementary Note 3.3 show that using different popular measures of diversity such as entropy does not affect the results on income diversity of encounters.

### Counterfactual simulation of mobility

To understand the underlying behavioral changes that contributed to the decrease of income diversity in urban encounters, we design a simulation framework that leverages the pre-pandemic data to create synthetic, counterfactual mobility patterns. The synthetic, counterfactual mobility dataset is designed so that while the fundamental behavioral patterns observed in 2019 are kept consistent, the number of users and stays at different place categories by different income quantiles are reduced to post-pandemic levels. This way, we are able to delineate the effects of different levels of behavioral changes on the total decrease in experienced income diversity.

The following steps are performed to simulate the synthetic mobility datasets. To create the synthetic counterfactual data for the year $y$ and month $m$, denoted as $\mathcal{S}_{y,m}$, we use the mobility data observed in the year 2019 on the same month $m$ as input data $\mathcal{D}_{2019,m}$, for example, to create a synthetic mobility dataset for April 2020, we use the mobility data observed in April 2019. Several different synthetic datasets, $\mathcal{S}_{y,m}^{(i)}$ and $\mathcal{S}_{y,m}^{(ii)}$ (and their variants), are created based on different levels of detail (see Supplementary Note 4). More specifically, the first synthetic dataset $\mathcal{S}_{y,m}^{(i)}$ is created by randomly removing visits from $\mathcal{D}_{2019,m}$ to adjust the total amount of dwell time spent at visits to places to match $\mathcal{D}_{y,m}$. The second synthetic dataset $\mathcal{S}_{y,m}^{(ii)}$ employs a more granular removal process, where we randomly remove visits to

places from $\mathcal{D}_{2019,m}$ by income quantiles $q$ and traveled distance $d$ (binned into 7 distance ranges: [0 km, 1 km), [1 km, 3 km), [3 km, 5 km), [5 km, 10 km), [10 km, 20 km), [20 km, 40 km), [40 km, ∞)) to adjust the amount of dwell time spent at visits to places. We also tested removing visits by income quantiles $q$, traveled distance $d$, and place taxonomy $c$, however, the results were similar to scenario (ii), as shown in Supplementary Note 4.2. More details on creating the counterfactual synthetic datasets can be found in Supplementary Note 4. For each counterfactual scenario, the simulations were run 10 times. After creating the synthetic counterfactual datasets, we compute the income diversity of encounters and compare with the experienced income diversity measured using the actual observed data $\mathcal{D}_{y,m}$ to delineate the effects of the reduction in active users and visits to place categories on the decrease in experienced income diversity. The income diversity metrics obtained from the 10 simulation runs were robust, with the average standard error of the estimated income diversity being 0.014% and 0.017% of the actual values of the place and individual income diversity, respectively. The standard errors are omitted from Fig. 2b since the confidence intervals are too small for visibility.

### Modeling the heterogeneity in experienced income diversity

To further understand how the income diversity of encounters decreased heterogeneously across sociodemographic groups throughout the pandemic, we build simple linear regression models of the form:

$$D_{CBG}(t), \Delta D_{CBG}(t) \sim \{R_{CBG}\} + \{P_{CBG}\} + \{M_{CBG}\} \qquad (1)$$

where $D_{CBG}(t)$ and $\Delta D_{CBG}(t)$ denote the differences in diversity at time $t$ compared to the same month in the year 2019. $\{R_{CBG}\}$ is the set of all residential variables from the census that describe the demographic, transportation, education, race, employment, wealth, etc. of the Census Block Group. $\{P_{CBG}\}$ is a vector of variables that indicate the places where individuals living in the CBG spent most of their time in 2019, out of the place subcategories that have at least 100 venues. For each individual, we identify the subcategories where the individual stays more than 0.3% of their time and obtain a binary vector with a length of 564, which is the number of place subcategories. $\{M_{CBG}\}$ is a set of variables that describe the geographical mobility behavior of people living in the corresponding CBG. We use two variables: (i) the radius of gyration of all the places visited by each user, and (ii) the average distance traveled to all places from each individual's home. Details of the regression covariates, including their summary statistics and correlations, are studied in Supplementary Note 5.

To further understand the differences in decreased experienced income diversity across CBSAs, we analyzed the correlation between the stringency of COVID-19 policies and the decrease in diversity. The stringency index $SI_{CBSA}(t)$ is a composite metric that measures the strictness of COVID-19 policies calculated using data collected in OxCGRT[46], and are provided at the state levels for the United States. The stringency index takes into account policies including the closings of schools and universities, closings of workplaces, canceling of public events and gatherings, the closing of public transport, orders to shelter-in-place, restrictions on internal movement between cities/regions and international travel, and presence of public info campaigns. More details are provided in the codebook in the GitHub webpage https://github.com/OxCGRT/covid-policy-tracker/blob/master/documentation/codebook.md.

### Reporting summary

Further information on research design is available in the Nature Portfolio Reporting Summary linked to this article.

## Data availability

The data that support the findings of this study are available from Spectus through their Social Impact program, but restrictions apply to

the availability of these data, which were used under the license for the current study and are therefore not publicly available. Information about how to request access to the data and its conditions and limitations can be found in https://spectus.ai/social-impact/. Data access requests should be submitted through Spectus' Social Impact customer page https://spectus.ai/lp/book-a-demo/, where the Sales team at Spectus may be contacted in a timely manner. Other data including the American Community Survey is available for download at https://data.census.gov/, and Tiger shapefiles can be downloaded from the US Census Bureau https://www.census.gov/programs-surveys/geography/guidance/tiger-data-products-guide.html.

## Code availability
The analysis was conducted using Python. Code to reproduce the main results in the figures from the aggregated data is publicly available on GitHub https://github.com/takayabe0505/IncomeDiversity.

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

## Acknowledgements

The authors would like to thank Spectus who kindly provided us with the mobility dataset for this research through their Data for Good program. E.M. acknowledges support by Ministerio de Ciencia e Innovación/Agencia Española de Investigación (MCIN/AEI/10.13039/501100011033) through grant PID2019-106811GB-C32, and from the National Science Foundation under Grant No. 2218748.

## Author contributions

T.Y. designed the algorithms, performed the analysis, and developed models and simulations. B.G.B.B. and E.M. performed part of the analysis, and partially developed models and simulations. A.P., X.D., and E.M. supervised the research. All authors wrote the paper. Company data were processed by T.Y. and partially by B.G.B.B. and E. M. All authors had access to aggregated (nonindividual) processed data. All authors reviewed the manuscript.

## Competing interests

The authors declare no competing interests.
