## [Peer Review File · Nature Communications]

Behavioral changes during the COVID-19 pandemic decreased income diversity of urban encountersREVIEWER COMMENTS

Reviewer #1 (Remarks to the Author):

The paper studies an important research topic: how the pandemic has reshaped the social lives in cities. The study is well-developed and the manuscript is overall well-written. The findings are important and inspiring for future studies. Overall, I am positive about the work. I do offer the following comments and suggestions for the authors to consider.

1. What is the place type of city & outdoor? Please explain and give examples.
2. Most results only provide location values (mean/median?). Please also provide ranges (confidence intervals, etc.). Some changes are not overly substantial, e.g., D_i in Figure 1C. It is unknown if the changes are significant or not.
3. When comparing D_a and D_i in Figure 1C, the former is much higher than the latter. So it seems that there is a certain mechanism in play that mitigate people's reducing diversity of encounters. It should be explained and explored.
4. Figure 2a "e.g., stay at home" seems to be a wrong example as the authors are reducing "outside activity."
5. It seems grocery stores are not, in fact, in 2020 April Fig. D. Please explain. Supermarkets also do not consistently show in three panels. The authors may also want to make the statement more accurate.
6. Which part of the counterfactual simulation (SI P19) is to examine changes in exploration behavior and place preferences? Is it ii-3? Please clarify how exploration and preferences have been changed here.

Reviewer #2 (Remarks to the Author):

This is a manuscript that investigates an important research topic. The manuscript is well-organized. The reviewer greatly appreciates the authors' efforts in the manuscript. However, there are several points that need to be improved and clarified. Thus, the reviewer recommends a major revision of the manuscript.

Major comments:

1. The reviewer encourages the authors to provide more in-depth discussions on recent studies on a similar topic. The authors state that "Despite the rich literature on the mobility dynamics and its impact on disease spread using location data collected via mobile phones, little is understood on how much longitudinal effects the pandemic has had on the quantity and quality of our encounters in urban environments." (Pages 1-2) However, the reviewer does not agree with this statement. Since the beginning of the pandemic, there have been many studies in the field of civil engineering, geography, and urban planning that addressed the longitudinal effects of the pandemic. These studies include (but are not limited to): Lee et al. (2020), Li et al. (2021), Huang et al. (2020, 2022), Kar et al. (2022), Kim & Kwan (2021), and Matson et al. (2021). Unfortunately, the manuscript does not provide a critical review of previous studies and does not highlight important gaps in those studies. Especially different impacts of the pandemic on people's mobility regarding their sociodemographic status have been extensively reported by those studies (e.g., Huang et al., 2022; Kar et al., 2022; Kim & Kwan, 2021). To sum up, the reviewer encourages the authors to review existing studies more comprehensively to highlight the added value and unique contribution of the manuscript.
Huang, X., Li, Z., Jiang, Y., Li, X., & Porter, D. (2020). Twitter reveals human mobility dynamics during the COVID-19 pandemic. *PloS one*, 15(11), e0241957.
Huang, X., Lu, J., Gao, S., Wang, S., Liu, Z., & Wei, H. (2022). Staying at home is a privilege: Evidence from fine-grained mobile phone location data in the United States during the COVID-19 pandemic. *Annals of the American Association of Geographers*, 112(1), 286-305.
Kar, A., Le, H. T., & Miller, H. J. (2022). What is essential travel? Socioeconomic differences in travel

demand in Columbus, Ohio, during the COVID-19 lockdown. *Annals of the American Association of Geographers*, 112(4), 1023-1046.

Kim, J., & Kwan, M. P. (2021). The impact of the COVID-19 pandemic on people's mobility: A longitudinal study of the US from March to September of 2020. *Journal of Transport Geography*, 93, 103039.

Lee, M., Zhao, J., Sun, Q., Pan, Y., Zhou, W., Xiong, C., & Zhang, L. (2020). Human mobility trends during the early stage of the COVID-19 pandemic in the United States. *PLoS One*, 15(11), e0241468.

Li, T., Wang, J., Huang, J., Yang, W., & Chen, Z. (2021). Exploring the dynamic impacts of COVID-19 on intercity travel in China. *Journal of Transport Geography*, 95, 103153.

Matson, G., McElroy, S., Lee, Y., & Circella, G. (2021). Longitudinal analysis of COVID-19 impacts on mobility: an early snapshot of the emerging changes in travel behavior. *Transportation Research Record*.

2. It is unclear why the authors particularly focused on four large cities - Boston, LA, Seattle, and Dallas. Cities in the US are diverse enough so that different patterns in travel behaviors and COVID-19 impacts are observed by previous studies. Considering that, the rationales for selecting those four cities are unclear, and they seem like arbitrary selections. Other (important) large cities, such as New York, Philadelphia, DC, and Chicago, are missing. Could the authors please address this issue?

3. It is unclear how mobility data can properly tell detailed place categories. The reviewer acknowledges that the authors incorporated POI data obtained from Foursquare. However, to the reviewer's best knowledge, mobile phone-based location (mobility) data are not able to tell detailed trip purposes. For example, if a user visits a mixed land-use area (e.g., commercial districts, such as the Prudential Center in Boston), it would be very difficult to know the specific place category (e.g., dining or shopping) that the person visited. The reviewer believes that the current mobile phone-based mobility data do not have the capacity to address this issue. Could the authors please clarify this?

4. Additionally, it is unclear how "the robustness of our results on the diversity of encounters" could be "tested using different spatial thresholds of d_{max} ." If the reviewer understands correctly, the authors utilized three threshold values. Why did the authors use these three specific values? Moreover, have these values been tested in other study areas in addition to Boston? Also, within a CBSA area, we observe a diverse range of land-use densities. For example, a smaller value can be proper for a high-dense area (e.g., downtown), but a larger value can be proper for a low-dense area, such as a suburb. How could the authors address this issue?

5. Appendix Section 4: The added value of counterfactual simulations to the manuscript is unclear. Especially, the following statement is difficult to understand: "while the fundamental behavioral patterns observed in 2019 are kept consistent, the number of visits to different place categories, in different distance ranges, by different income quantiles are reduced to post-pandemic levels. This way, we are able to delineate the effects of different levels of behavioral changes to the total decrease in income diversity." It is difficult to understand how the authors have the capacity to delineate "the effects of different levels of behavioral changes" by using the pre-pandemic data. This sounds like the authors want to differentiate changes in mobility that would be made without considering the COVID-19 pandemic. However, the reviewer is skeptical about the scientific soundness of this simulation, given the limited data. In other words, since the pandemic affects the entire world, how can the authors differentiate the control group and the experimental group to identify specific causes of mobility changes? Could the authors please address this issue?

Minor comments:

1. Appendix Section 1.3: To the reviewer's best knowledge, Infogroup POI data can be more comprehensive than Foursquare data. This is because Foursquare data are based on users' voluntary input, which may lead to uncertainties and inaccuracy. Infogroup POI data seem more robust and thus are widely utilized by researchers. Why did the authors need to utilize Foursquare data rather than

Infogroup data?

2. Appendix Section 1.4: The reviewer appreciates the authors' efforts to conduct many sensitivity tests to ensure their analytical robustness. However, it is unclear why we need to conduct robustness tests regarding the number of income quantiles. From the practical perspective, researchers can go with 5/7/10 quartiles without providing justification because they provide more meaningful information than 2 or 3 quartiles. If the authors pertain to keeping robust test results, the reviewer suggests that presenting 4-6 quartiles results would be proper.

3. Appendix Section 1.4: Additionally, please explain which census data (e.g., American Community Survey or Decennial Census) are used for income. Was the median income level used? If the ACS data are utilized, the authors might need to consider (or at least discuss) spatial uncertainties (Jung et al., 2019). Also, there is a discrepancy between the analytical unit between mobility data (individual-level) and income-level data (census block group level). How could the authors address this issue?
Jung, P. H., Thill, J. C., & Issel, M. (2019). Spatial autocorrelation and data uncertainty in the American Community Survey: a critique. *International Journal of Geographical Information Science*, 33(6), 1155-1175.

4. Results "Using a large and longitudinal dataset of individual GPS location records ..." (Page 2): This statement can be misleading. Although the data is originally based on individual GPS location records, the actual analysis is based on the aggregated data, which is substantially different from the individual GPS location records. Please consider editing this statement.

5. Figure 1A: This map can be improved cartographically. This map cannot provide meaningful information for those who are not familiar with Boston. North arrow and labels of important landmarks (e.g., Charles River, Downtown, etc.) would be helpful. This comment can also be applied to Figure 3A. Please consider updating Figure 3A as well.

6. "Behavioral changes worsened income diversity in cities": Throughout the manuscript, the term "income diversity" is utilized frequently. It seems that this term means the level of diversity of people's income level who visited a certain place (POI). However, the term "income diversity" can be confused with income inequality related to Gini's coefficient. The reviewer suggests that the authors come up with a clearer term than the term "income diversity." Also, there are many existing studies on segregation that people experience in their daily life (e.g., Palmer, 2013; Park & Kwan, 2018; Wang et al., 2013). How is the term "income diversity" conceptually different from these studies? Could the authors please address this issue?

Wang, D., Li, F., & Chai, Y. (2012). Activity spaces and sociospatial segregation in Beijing. *Urban Geography*, 33(2), 256-277.

Palmer, J. R. (2013). Activity-space segregation: Understanding social divisions in space and time (Doctoral dissertation, Princeton University).

Park, Y. M., & Kwan, M. P. (2018). Beyond residential segregation: A spatiotemporal approach to examining multi-contextual segregation. *Computers, Environment and Urban Systems*, 71, 98-108.

7. The reviewer suggests that the authors discuss the limitations of mobile phone-based mobility data. For example, algorithmic uncertainty related can be an important limitation (Kwan, 2016; Mooney & Pejaver, 2018). Please discuss this important limitation that might affect the results of the study. In other words, the reviewer argues that important limitations of mobile phone-based mobility data should be discussed explicitly despite its many benefits.

Kwan, M. P. (2016). Algorithmic geographies: Big data, algorithmic uncertainty, and the production of geographic knowledge. *Annals of the American Association of Geographers*, 106(2), 274-282.

Mooney, S. J., & Pejaver, V. (2018). Big data in public health: terminology, machine learning, and privacy. *Annual Review of Public Health*, 39, 95.

8. Discussion "Our results suggest that policy interventions on urban mobility, such as the introduction

of fare-free transit systems and development of public spaces, should target and evaluate the recovery of social exploration to potentially improve income diversity after the pandemic." (Page 8): It is unclear how this argument is supported by findings from the current manuscript's results. If certain groups of people experience limited daily mobility and thus result in isolation, increasing mobility can be a meaningful policy intervention. However, enhancing people's mobility just to "improve income diversity" can question its rationales for policy implementation. Please clarify this issue. Additionally, another statement - "Increasing the quantity and diversity of our social encounter could help communities to acquire social capital, which could improve the resilience to natural hazards and foster economic growth." - is questionable and needs more justifications. To sum up, the reviewer argues that the policy implications mentioned in the manuscript might not be based on the primary results of this manuscript.

Reviewer #3 (Remarks to the Author):

The authors of this paper add to the extant literature on the impact of COVID-19 on human behavior, specifically analyzing changes in mobility patterns in 4 large metropolitan areas by means of a massive dataset of anonymized individuals provided by a third party. The paper is well-written, the methodology is sound, the experiments are carried out in a rigorous fashion and the results are of general interest and impact. I praise the authors for providing unnumbered robustness checks and additional material to motivate their main findings, and believe this paper is worthy of publication in Nature Communications. However, I believe that it also needs some revision work which I am sure can be easily addressed by the authors. I will try to list my concerns below following the order of the manuscript

Major:

- Similar to several papers published in NC (as far as I've seen), I would add a few sentences on motivation/gaps in the literature/results/conclusions at the end of the introduction, to provide a general idea of the paper to the reader
- I recommend the author to review captions of figures: they should first describe the content shown and then make conclusions, without requiring the reader to go back and forth from the text to the figures. Also, I would try to be more precise in the Supplementary: many captions do not clearly tell what they show.
- I recommend the author to carry out statistical tests (whenever it's feasible) when claiming differences.
- The first of the Results subsections is more of a Methods section, and I would move it from there (or remove it if the authors think that the current Methods provide what is needed). I would still try to define the diversity variable.
- (Result 1) The authors provide results for the average value of several metrics; could they show distributions via boxplots (in the Supplementary maybe) and/or motivate the choice of such statistic? How about the median (if it makes sense)?
- (Result 1) It is not clear whether the monthly observations shown in Figure 1 are computed as (1) the difference of the mean of the distributions (of individuals' values) for month 2019 and month X or (2) as the mean value of the individuals' differences (month 2019 - month X). Please specify and motivate.
- (Figure 1) It contains a mixture of Boston-specific results and results for other cities, and it might be confusing to the reader; I would try to separate them (maybe two separate rows?).
- (Figure 1) I wonder whether panel A could better convey its message if plotted like the one in Figure 3, i.e. showing the difference w.r.t the baseline as well as the distribution.
- (Figure 1) I don't quite like panel D: would it be still clear if plotted like Figure S8, i.e. with all monthly observations and not just 3? Alternatively, if authors want to show those 3 months specifically, maybe a heatmap could be fancier (x-axis: categories, y-axis: month, cell: value of the delta, with a colorbar)
- (Result 2): the introductory paragraph is confusing: there is an "In addition" which gives the idea that the scenarios are 4 rather than 3. Please clarify.

- (Result 2): If I understand it correctly, random counterfactual datasets are simulated only once. I would expect instead N simulations and then compute the average difference with the empirical data, but I might be missing something here.
 - (Result 2): "reduction in total activities caused around 50% of the decrease in diversity" this sentence is confusing, because the figure shows the absolute decrease of diversity, so one should see that out of -30%, -15% is accounted for by the reduction in total activities. Could the authors show the composition of the reduction – which is shown in Figure 1 – instead, alongside with the absolute values of reduction?
 - (Result 2): Are results shown in Figure S14 statistically significant ("Although we observe different rates of dwell time decrease and recovery across income quantiles where the richer populations disproportionately reduce dwell times at places than poorer populations")? I believe it shows only the average values of POIs for each quantile, if I understand it correctly.
 - (Figure 2): I appreciate it but I don't particularly like panel A, I wonder whether it really adds much to the understanding of the results.
 - (Figure 2): How are colors assigned to categories in panel D? Please clarify.
 - (Result 3): I am not very convinced by the approach used by the authors to find 'predictors' of variation in diversity, namely building a regression model for each month. I am not an expert of econometric models but I think this part needs some further motivation and/or alternative analyses. I understand that Panel regressions or ARIMA are not suitable because the socio-economic/demographic variables are fixed, but maybe carrying out a regression with all the monthly observations at once could be one direction. Also, some descriptive analyses could strengthen their findings, e.g., show that if variable X is positively associated with variation, then the top percentiles in the distribution of X should consistently exhibit a larger variation over time. Worst case, the authors could find some strong references where the same approach is employed on similar data/analyses.
 - (Figure 3): In light of my comments above, I am a bit skeptical about panels B and C.
 - (Result 4): Authors say "we tested ARIMA type models as well, however the regression results and especially the estimated coefficients were found to be robust" but a double check with Table S14 does not show this (cf. Los Angeles and Dallas). Please clarify.
 - (Conclusions): I think the authors expand a bit more on the limitations of their study.
- Minor:
- Can the authors motivate the choice of the four cities (although I believe it's more a matter of data availability)?
 - I recommend adding a few pointers to the literature related to the (broad) socio-economic effects of the pandemic (including NPIs) inferred via mobility data as well as the epidemiological literature on the effects of mobility restrictions. I will not provide specific references to avoid biases, and I am sure the authors can find appropriate citations
 - I recommend ensuring more consistency between the manuscript and the supplementary material, i.e., please make sure to reference figures and tables in the order they are mentioned in the main text, as much as possible.
 - I recommend making another round of proof-reading: there are a few typos starting from the abstract (e.g. inconsistencies of verbs with singular/plural/third person) and some sentences sound a bit wordy (e.g. "this agrees with ..")
 - The GitHub repository cannot be accessed (and figures cannot be reproduced) , but I doubt this happened on purpose. Please put it back online.

Responses to the Reviewers for Behavioral changes during the pandemic worsened income diversity of urban encounters

Takahiro Yabe, Bernardo Garcia Bulle Bueno, Xiaowen Dong,
Alex ‘Sandy’ Pentland, and Esteban Moro

We thank the reviewers for their helpful comments, questions and suggestions and their thorough review of our manuscript. We have extensively revised the manuscript based on the provided reviews to address their concerns. We list the revisions made and responses to each comment provided by the reviewers below.

Reviewer #1:

The paper studies an important research topic: how the pandemic has reshaped the social lives in cities. The study is well-developed and the manuscript is overall well-written. The findings are important and inspiring for future studies. Overall, I am positive about the work. I do offer the following comments and suggestions for the authors to consider.

We thank the referees for the positive and constructive comments.

1. What is the place type of city & outdoor? Please explain and give examples.

The “City & Outdoor” category contains parks (e.g., Boston Common), public spaces, fields, beaches, and historic sites. We have added this description to the text in Supplementary Material Page 8, Line 174.

2. Most results only provide location values (mean/median?). Please also provide ranges (confidence intervals, etc.). Some changes are not overly substantial, e.g., D_i in Figure 1C. It is unknown if the changes are significant or not.

Thank you for your suggestion. We have omitted error bars since they were negligible and are not visible due to the large number of POIs (50K~200K depending on the city) and users (140K~450K). On average, the standard error of the average individual and place segregation metrics are 0.13% and 0.14% of the actual values, respectively. Therefore, the changes shown in Figure 1C are statistically significant. We have added this point to the caption of Figure 1, and also added the standard error bars in Supplementary Figure S12.

3. When comparing D_a and D_i in Figure 1C, the former is much higher than the latter. So it seems that there is a certain mechanism in play that mitigate people’s reducing diversity of encounters. It should be explained and explored.

D_i is a convoluted version of the D_α for all places α visited by i , which considers the ratio of the sum of stay durations across all income quantiles. Although each place α could significantly lose income diversity during the pandemic due to decrease in visits, individuals are likely to visit multiple places in a given month, which results in a higher D_i . We have added this discussion in Page 4 Lines 104: “ $D_{\{i\}}$ is a convoluted version of the $D_{\{\alpha\}}$ for all places α visited by i , which considers the ratio of the sum of stay duration across all income quantiles. Thus, although each place α could significantly lose income diversity during the pandemic due to decrease in visits, individuals are likely to visit multiple places in a given month, which results in a higher $D_{\{i\}}$ than $D_{\{\alpha\}}$.”.

4. Figure 2a “e.g., stay at home” seems to be a wrong example as the authors are reducing "outside activity."

We mentioned “stay at home” as a paraphrase for “reduction of total activity”, not an example of an “activity”. We agree that this was confusing, and have removed that annotation.

5. It seems grocery stores are not, in fact, in 2020 April Fig. D. Please explain. Supermarkets also do not consistently show in three panels. The authors may also want to make the statement more accurate.

Thank you for pointing this out. We have made the explanations more accurate in Page 5 Lines 159 – 164: “Supplementary Figure S24 shows the POI sub-categories which were more (and less) visited in different periods during the pandemic compared to 2019 levels. Hardware stores, big box stores, grocery stores (in October 2020 and 2021) were POI sub-categories which gained popularity during the pandemic, and gyms, movie theaters, American food places were subcategories which were less visited frequently. ”

6. Which part of the counterfactual simulation (SI P19) is to examine changes in exploration behavior and place preferences? Is it ii-3? Please clarify how exploration and preferences have been changed here.

Thank you for your question. Given that the decrease in income diversity is not fully explained by the reduction in mobility (scenario i), travel distance reduction and differences between income quantiles (scenario ii), or reduction in visits to major place categories (scenario ii-3 in Supplementary Material), the remainder of the decreased income diversity can be associated with changes in behavioral patterns. To investigate what aspects of human mobility behavior have changed during the pandemic, we examine how parameters of the Social-EPR model have changed. Out of all of the model parameters that govern the mobility behavior, social exploration and place preferences were mobility characteristics that had the most significant changes during the pandemic, hence are shown in Figure 2. We also tested that the decrease in social exploration during the pandemic was excessive compared to the level of decrease that would be expected in counterfactual scenario (ii), as shown in Supplementary Figure S23.

Reviewer #2:

This is a manuscript that investigates an important research topic. The manuscript is well-organized. The reviewer greatly appreciates the authors' efforts in the manuscript. However, there are several points that need to be improved and clarified. Thus, the reviewer recommends a major revision of the manuscript.

Major comments:

1. The reviewer encourages the authors to provide more in-depth discussions on recent studies on a similar topic. The authors state that "Despite the rich literature on the mobility dynamics and its impact on disease spread using location data collected via mobile phones, little is understood on how much longitudinal effects the pandemic has had on the quantity and quality of our encounters in urban environments." (Pages 1-2) However, the reviewer does not agree with this statement. Since the beginning of the pandemic, there have been many studies in the field of civil engineering, geography, and urban planning that addressed the longitudinal effects of the pandemic. These studies include (but are not limited to): Lee et al. (2020), Li et al. (2021), Huang et al. (2020, 2022), Kar et al. (2022), Kim & Kwan (2021), and Matson et al. (2021). Unfortunately, the manuscript does not provide a critical review of previous studies and does not highlight important gaps in those studies.

Especially different impacts of the pandemic on people's mobility regarding their sociodemographic status have been extensively reported by those studies (e.g., Huang et al., 2022; Kar et al., 2022; Kim & Kwan, 2021). To sum up, the reviewer encourages the authors to review existing studies more comprehensively to highlight the added value and unique contribution of the manuscript.

Huang, X., Li, Z., Jiang, Y., Li, X., & Porter, D. (2020). Twitter reveals human mobility dynamics during the COVID-19 pandemic. *PloS one*, 15(11), e0241957.

Huang, X., Lu, J., Gao, S., Wang, S., Liu, Z., & Wei, H. (2022). Staying at home is a privilege: Evidence from fine-grained mobile phone location data in the United States during the COVID-19 pandemic. *Annals of the American Association of Geographers*, 112(1), 286-305.

Kar, A., Le, H. T., & Miller, H. J. (2022). What is essential travel? Socioeconomic differences in travel demand in Columbus, Ohio, during the COVID-19 lockdown. *Annals of the American Association of Geographers*, 112(4), 1023-1046.

Kim, J., & Kwan, M. P. (2021). The impact of the COVID-19 pandemic on people's mobility: A longitudinal study of the US from March to September of 2020. *Journal of Transport Geography*, 93, 103039.

Lee, M., Zhao, J., Sun, Q., Pan, Y., Zhou, W., Xiong, C., & Zhang, L. (2020). Human mobility trends during the early stage of the COVID-19 pandemic in the United States. *PLoS One*, 15(11), e0241468.

Li, T., Wang, J., Huang, J., Yang, W., & Chen, Z. (2021). Exploring the dynamic impacts of COVID-19 on intercity travel in China. *Journal of Transport Geography*, 95, 103153.

Matson, G., McElroy, S., Lee, Y., & Circella, G. (2021). Longitudinal analysis of COVID-19 impacts on mobility: an early snapshot of the emerging changes in travel behavior. *Transportation Research Record*.

Thank you for your suggestions. By “Despite the rich literature on the mobility dynamics and its impact on disease spread using location data collected via mobile phones, little is understood on how much longitudinal effects the pandemic has had on the quantity and quality of our encounters in urban environments.”, we intended to point out that little is known on the longitudinal (i.e., 3 years after the beginning of the pandemic) effects. Most of the bibliography suggested above studied the initial impact of the pandemic on mobility patterns, especially only during the first wave. We have made this point clearer in the manuscript. We do agree that we have missed some important publications in this field, and so we have substantially revised the introduction and included the references you have shared with us.

Page 2, Line 36 now reads: “There have been many studies in the fields of civil engineering, geography, and urban planning that have analyzed the short-term mobility dynamics during the early stages of the pandemic and lockdown [30-33], socioeconomic disparities in response to lockdowns [34-36], and its impact on disease spread using location data collected via mobile phones [37,38].”

2. It is unclear why the authors particularly focused on four large cities - Boston, LA, Seattle, and Dallas. Cities in the US are diverse enough so that different patterns in travel behaviors and COVID-19 impacts are observed by previous studies. Considering that, the rationales for selecting those four cities are unclear, and they seem like arbitrary selections. Other (important) large cities, such as New York, Philadelphia, DC, and Chicago, are missing. Could the authors please address this issue?

The limited number of cities focused in the study is mainly due to limitations in data access and computational resources. Strict limitation to data access is imposed by our data providers, to ensure ethical and legal use of the data in a privacy-preserving platform. Given these constraints, the four cities were selected to maximize the diversity of characteristics in terms of geographical locations, sociodemographic details, political inclinations, weather characteristics, COVID-19 policy strictness, etc.

3. It is unclear how mobility data can properly tell detailed place categories. The reviewer acknowledges that the authors incorporated POI data obtained from Foursquare. However, to the reviewer's best knowledge, mobile phone-based location (mobility) data are not able to tell detailed trip purposes. For example, if a user visits a mixed land-use area (e.g., commercial districts, such as the Prudential Center in Boston), it would be very difficult to know the specific place category (e.g., dining or shopping) that the person visited. The reviewer believes that the current mobile phone-based mobility data do not have the capacity to address this issue. Could the authors please clarify this?

We thank your comment on the accuracy of the dataset. In our paper we use industry-grade GPS data from location-based services, which is used by private companies, governments and agencies to detect footfall at store level. GPS data is rather accurate, with errors around 5-10 meters in most cases in urban areas. That precision allows us to distinguish people's visitations to individual places, including, for example, different stores in a shopping mall or other commercial areas. GPS data has been successfully used during the pandemic to understand the effect of non-pharmaceutical interventions in visits to different places, or even to approximate the contact network between people in different settings, see for example, Chang, S. *et al.*

Mobility network models of COVID-19 explain inequities and inform reopening. *Nature* 589, 82–87 (2021) or our previous works, references [19] and [38].

In our work, we find that the average spatial error of a collection of observations during a stay is 8.31 meters, which allows us to detect the specific place (and its category) where the individual is staying. To that end, we match the stays to POI data using a simple distance-based attribution method. However, we have extensively checked that our results do not depend on the details of the attribution method (see next question).

4. Additionally, it is unclear how "the robustness of our results on the diversity of encounters" could be "tested using different spatial thresholds of d_{\max} ." If the reviewer understands correctly, the authors utilized three threshold values. Why did the authors use these three specific values? Moreover, have these values been tested in other study areas in addition to Boston?

Also, within a CBSA area, we observe a diverse range of land-use densities. For example, a smaller value can be proper for a high-dense area (e.g., downtown), but a larger value can be proper for a low-dense area, such as a suburb. How could the authors address this issue?

We have tested different spatial thresholds (50m, 100m, 150m, 200m) for visit attribution to POIs to test the sensitivity of the algorithm. These threshold values were picked as adequate values within a reasonable spatial range with respect to distances between closest POIs. For reference, the diameter of the smallest census block group in Boston MSA (in Back Bay) is around 120 meters. We have added the same tests for the other three cities in Supplementary Figure S1.

These values are maximum distances in which a stay location may be attributed to the closest POI and thus they are upper bounds. In reality, the median distance to the attributed POI is much smaller, around 23.5 meters (mode of distribution is 10.0 meters). The point you mention is not an issue, since the visit attribution algorithm chooses the closest POI from the stay location. Thus, in areas with large numbers of POIs (like downtown), the effective distance to the attributed closest POI is even smaller. Thus, effectively, our attribution method adapts to areas with different density of POIs. We have included a discussion about this effective adaptive mechanism in the Section 1.2 in the Supplementary Material. We have also included references to similar studies by the same group in which those robustness checks were implemented on the same data for pre-pandemic income experienced diversity or even for epidemiological studies of the transmission of COVID-19, references [19] and [38] in the main paper. In all cases, it was found that our results were robust towards those thresholds.

5. Appendix Section 4: The added value of counterfactual simulations to the manuscript is unclear. Especially, the following statement is difficult to understand: "while the fundamental behavioral patterns observed in 2019 are kept consistent, the number of visits to different place categories, in different distance ranges, by different income quantiles are reduced to post-pandemic levels. This way, we are able to delineate the effects of different levels of behavioral changes to the total decrease in income diversity." It is difficult to understand how the authors have the capacity to delineate "the effects of different levels of behavioral changes" by using the pre-pandemic data. This sounds like the authors want to differentiate changes in mobility that would be made without considering the COVID-19 pandemic. However, the reviewer is skeptical about the scientific soundness of this simulation, given the limited data. In other words,

since the pandemic affects the entire world, how can the authors differentiate the control group and the experimental group to identify specific causes of mobility changes? Could the authors please address this issue?

We would like to clarify how we set up the counterfactual experiment.

The aim of the counterfactual scenario (i) is to understand ‘how would the income diversity change if people just simply decreased the number of visits to places from pre-pandemic levels?’ This is an important question since one could hypothesize that the income diversity decreased in 2020 just because the number of encounters decreased at places (leading to a sparser count of encounters). To test the effects of the reduction of visits, we randomly removed visits from the data in 2019 to match the number of visits to the 2020 level. We can then calculate the income diversity of the ‘reduced 2019 dataset’ and ‘2020 dataset’, and compare how much of the diversity decrease is due to decrease in visits. In the paper, we show that the decrease in visits was associated with around 50% of the diversity decrease in the early stages of the pandemic, but later disappeared in late 2021 since the number of visits to places had recovered back to pre-pandemic levels.

Scenario (ii) and its variants perform the same operation, but reduces the number of visits by different place categories, distance ranges, and income quantiles. This way, we are able to test whether the decrease in diversity was due to drops in visits to specific place categories (e.g., schools), or visits to specific distance ranges (e.g., long distance trips), or visits by specific income quantile groups (e.g., richest quantile). We have further clarified this in the main manuscript, Page 4 Lines 129-134.

Minor comments:

1. Appendix Section 1.3: To the reviewer's best knowledge, Infogroup POI data can be more comprehensive than Foursquare data. This is because Foursquare data are based on users' voluntary input, which may lead to uncertainties and inaccuracy. Infogroup POI data seem more robust and thus are widely utilized by researchers. Why did the authors need to utilize Foursquare data rather than Infogroup data?

We thank your comments and suggestions on the POI dataset. The Infogroup (now Data Axle) data contains the geolocation of the address of millions of businesses in the US. Although very valuable, their data is mainly restricted to business and commercial activities. Thus, it does not cover other important places important to understand the mobility changes during COVID-19 pandemic, such as visits to outdoor places. More importantly, the geolocation of Infogroup places is recorded at the official address of the business., Thus there are instances where different places are clustered into the exact same location if located in the same shopping mall. Sometimes the location of the official address of the mall is at the gate, located far away from any POI.

On the other hand, Foursquare data contains location data of different types of POIs such as parks, and also contains the precise location coordinates of the POIs, which is why we have proceeded to use the Foursquare POI as the main dataset. For example, in Quincy Market in Boston, which is one of the most attractive shopping and dining spots for tourists in the city, Infogroup data has 1 location coordinate containing information of 20 points of interest, while in the Foursquare data, each place is assigned a unique coordinate within the shopping mall. We also note that Foursquare data does not only come from voluntary input. The

data is built from a combination of crowd-sourced user activity and the aggregation of data from additional sources and has third-party verification (Foursquare, 2020). Finally, a 2018 study comparing the Foursquare POI database with other public POI databases from mapping and social media platforms (Facebook, Foursquare, Google, Instagram, Open Street Map, Twitter, and Yelp) established that while none of these databases is complete, Foursquare's data quality, as measured by number of POI, number of categories included, and positioning accuracy, was among the best (Hochmair, 2018).

We have added a discussion about these points in the Supplementary Material Section 1.3.

2. Appendix Section 1.4: The reviewer appreciates the authors' efforts to conduct many sensitivity tests to ensure their analytical robustness. However, it is unclear why we need to conduct robustness tests regarding the number of income quantiles. From the practical perspective, researchers can go with 5/7/10 quantiles without providing justification because they provide more meaningful information than 2 or 3 quantiles. If the authors pertain to keeping robust test results, the reviewer suggests that presenting 4-6 quantiles results would be proper.

Thank you for your suggestion. We agree with your point, but also believe it is still meaningful to test whether using 2 or 3 income quantiles yield similar robust results on income diversity, so we have kept the result as it is. Moreover, some traditional inequality or segregation metrics (like poor-rich exposure, see <https://www.census.gov/topics/housing/housing-patterns/guidance/appendix-b.html>) are based on only 2 quantiles (rich vs. poor), so we wanted to understand if our results were still valid under such strong division of people into two groups.

3. Appendix Section 1.4: Additionally, please explain which census data (e.g., American Community Survey or Decennial Census) are used for income. Was the median income level used? If the ACS data are utilized, the authors might need to consider (or at least discuss) spatial uncertainties (Jung et al., 2019). Also, there is a discrepancy between the analytical unit between mobility data (individual-level) and income-level data (census block group level). How could the authors address this issue?

Jung, P. H., Thill, J. C., & Issel, M. (2019). Spatial autocorrelation and data uncertainty in the American Community Survey: a critique. *International Journal of Geographical Information Science*, 33(6), 1155-1175.

We thank the reviewer for bringing up the potential problem of uncertainty in the ACS or other census data. We have included details on which census data we used for income estimates: "2016-2020 5-year American Community Survey (ACS)". Note that in our case, we used the median household income from the ACS as a proxy of socio-economic status and to determine the broad income quantiles the users belong to. Thus, we are not using the exact median household income, but a general classification of users into four quantile groups. Thus, our income classification is only slightly affected by uncertainty problems in the actual ACS estimates, see Spielman, S. E. & Singleton, A. Studying Neighborhoods Using Uncertain Data from the American Community Survey: A Contextual Approach. *Ann Assoc Am Geogr* 105, 1003–1025 (2015). We have added a discussion about this point in the Supplementary Material Section 1.4. The difference in the analytical unit between mobility data (individual level) and income data (census block group level) is a minor issue for this study, since our aim was not to conduct an individual level study (i.e., we are not testing

diversity changes at the individual level). Rather, we are conducting analysis of the income diversity changes on the census block group level (or on the CBSA level for Figures 1, 2, and 4).

4. Results "Using a large and longitudinal dataset of individual GPS location records ..." (Page 2): This statement can be misleading. Although the data is originally based on individual GPS location records, the actual analysis is based on the aggregated data, which is substantially different from the individual GPS location records. Please consider editing this statement.

Thank you for your suggestion. Although the analysis results are mainly presented in the aggregate level (CBG level or CBSA level), the original analysis is conducted using individual GPS location records. To avoid confusion, we have removed the word "individual", and now state that we "used a large and longitudinal dataset of GPS location records". For reference, there have been many papers that use individual mobility data and present aggregated results that state that they have used individual records, such as the following:

- Lucchini, L., Centellegher, S., Pappalardo, L., Gallotti, R., Privitera, F., Lepri, B., & De Nadai, M. (2021). Living in a pandemic: changes in mobility routines, social activity and adherence to COVID-19 protective measures. *Scientific Reports*, 11(1), 1-12.
- Moro, E., Calacci, D., Dong, X., & Pentland, A. (2021). Mobility patterns are associated with experienced income segregation in large US cities. *Nature Communications*, 12(1), 1-10.
- Schneider, C. M., Belik, V., Couronné, T., Smoreda, Z., & González, M. C. (2013). Unravelling daily human mobility motifs. *Journal of The Royal Society Interface*, 10(84), 20130246.

5. Figure 1A: This map can be improved cartographically. This map cannot provide meaningful information for those who are not familiar with Boston. North arrow and labels of important landmarks (e.g., Charles River, Downtown, etc.) would be helpful. This comment can also be applied to Figure 3A. Please consider updating Figure 3A as well.

Thank you for the suggestion. We have updated Figures 1A and 3A with labels of important landmarks and North arrows.

6. "Behavioral changes worsened income diversity in cities": Throughout the manuscript, the term "income diversity" is utilized frequently. It seems that this term means the level of diversity of people's income level who visited a certain place (POI). However, the term "income diversity" can be confused with income inequality related to Gini's coefficient. The reviewer suggests that the authors come up with a clearer term than the term "income diversity."

Also, there are many existing studies on segregation that people experience in their daily life (e.g., Palmer, 2013; Park & Kwan, 2018; Wang et al., 2013). How is the term "income diversity" conceptually different from these studies? Could the authors please address this issue?

Wang, D., Li, F., & Chai, Y. (2012). Activity spaces and sociospatial segregation in Beijing. *Urban Geography*, 33(2), 256-277.

Palmer, J. R. (2013). Activity-space segregation: Understanding social divisions in space and time (Doctoral dissertation, Princeton University).

Park, Y. M., & Kwan, M. P. (2018). Beyond residential segregation: A spatiotemporal approach to examining multi-contextual segregation. *Computers, Environment and Urban Systems*, 71, 98-108.

Thank you for your suggestion. We think that the meaning of the term ‘income diversity’ is clear when stated ‘income diversity of (urban) encounters’, however, we agree that the phrase ‘income diversity’ may be confused with other terms such as income inequality. Therefore, we decided to use the phrase ‘experienced income diversity’ when not coupled with the word ‘encounters’ to make the term clearer.

The papers listed here have similar objectives with our study, and although specifics of the mathematical formulation of diversity (inverse of segregation) are slightly different, they are conceptually similar, and so we have added them to the Introduction. However, the aim and contribution of our study was not to develop a new metric for measuring the diversity of encounters, but to provide empirical research using large scale mobility data (which these studies do not use) and human mobility models into how and why people and places have become less diverse during the 3-year period during the pandemic.

7. The reviewer suggests that the authors discuss the limitations of mobile phone-based mobility data. For example, algorithmic uncertainty related can be an important limitation (Kwan, 2016; Mooney & Pejaver, 2018). Please discuss this important limitation that might affect the results of the study. In other words, the reviewer argues that important limitations of mobile phone-based mobility data should be discussed explicitly despite its many benefits.

Kwan, M. P. (2016). Algorithmic geographies: Big data, algorithmic uncertainty, and the production of geographic knowledge. *Annals of the American Association of Geographers*, 106(2), 274-282.

Mooney, S. J., & Pejaver, V. (2018). Big data in public health: terminology, machine learning, and privacy. *Annual Review of Public Health*, 39, 95.

Thank you for this suggestion. Although we have paid careful attention to potential biases and limitations of mobile phone location data using various robustness checks and post-stratification analysis regarding the data, we have added this to our discussion. Page 8 Line 253 reads: “Although we have tested the robustness of the used mobile phone data from various aspects and applied a post-stratification strategy to overcome sampling biases, there may be overlooked biases arising from uncertainty in data collection algorithms, such as the frequency and timing of location data collection [54,55].”

8. Discussion "Our results suggest that policy interventions on urban mobility, such as the introduction of fare-free transit systems and development of public spaces, should target and evaluate the recovery of social exploration to potentially improve income diversity after the pandemic." (Page 8): It is unclear how this argument is supported by findings from the current manuscript's results. If certain groups of people experience limited daily mobility and thus result in isolation, increasing mobility can be a meaningful policy intervention. However, enhancing people's mobility just to "improve income diversity" can question its rationales for policy implementation. Please clarify this issue.

Additionally, another statement - "Increasing the quantity and diversity of our social encounter could help communities to acquire social capital, which could improve the resilience to natural hazards and foster economic growth." - is questionable and needs more justifications. To sum up, the reviewer argues that the

policy implications mentioned in the manuscript might not be based on the primary results of this manuscript.

Thank you for your questions and suggestions. We admit that the statement lacked clarity. We have edited the corresponding sentences in the discussion. Page 8 Line 247 reads: “Our results suggest that policy interventions on urban mobility that enable populations of different income groups to increase physical encounters should also target and evaluate the recovery of social exploration to potentially improve experienced income diversity after the pandemic. Other studies have recently highlighted the importance of these weak ties on career success [Rajkumar et al., *Science* 2022] and economic mobility [Chetty et al., *Nature* 2022]. Such public interventions include the introduction of fare-free transit systems and development of public spaces that aim to lower the cost of transport and opportunity from low-income areas to the high-income areas.”

Reviewer #3:

The authors of this paper add to the extant literature on the impact of COVID-19 on human behavior, specifically analyzing changes in mobility patterns in 4 large metropolitan areas by means of a massive dataset of anonymized individuals provided by a third party. The paper is well-written, the methodology is sound, the experiments are carried out in a rigorous fashion and the results are of general interest and impact. I praise the authors for providing unnumbered robustness checks and additional material to motivate their main findings, and believe this paper is worthy of publication in Nature Communications. However, I believe that it also needs some revision work which I am sure can be easily addressed by the authors. I will try to list my concerns below following the order of the manuscript

Major:

1. Similar to several papers published in NC (as far as I've seen), I would add a few sentences on motivation/gaps in the literature/results/conclusions at the end of the introduction, to provide a general idea of the paper to the reader

Thank you for your suggestion. We have added a few sentences in the introduction. Page 2 Line 42 ‘To this end, this study contributes to bridging this knowledge gap by analyzing how the income diversity of urban encounters have changed before and during the pandemic across a 3-year timespan. We further unravel the mobility behavior changes that contributed to the decreased diversity, using mobility data from 4 major US cities and human mobility models.’

2. I recommend the author to review captions of figures: they should first describe the content shown and then make conclusions, without requiring the reader to go back and forth from the text to the figures. Also, I would try to be more precise in the Supplementary: many captions do not clearly tell what they show.

Thank you for your suggestions. We have substantially reviewed the captions of the figures both in the main manuscript and the supplementary material.

3. I recommend the author to carry out statistical tests (whenever it's feasible) when claiming differences.

Thank you for your suggestion. We have added statistical tests whenever we claim differences. In terms of differences in the mean income diversity values (both for individuals and places), the differences are statistically significant via Welch's t-tests due to the large number of POIs (50K~200K depending on the city) and users (140K~450K). On average, the standard error of the average individual and place segregation metrics are 0.13% and 0.14% of the actual values, respectively. We have added this point to the captions in Figure 1C.

4. The first of the Results subsections is more of a Methods section, and I would move it from there (or remove it if the authors think that the current Methods provide what is needed). I would still try to define the diversity variable.

Thank you for your suggestion. We believe the current level of detail provided in the first of the results subsections is sufficient and necessary for readers to understand how we are measuring income diversity.

5. (Result 1) The authors provide results for the average value of several metrics; could they show distributions via boxplots (in the Supplementary maybe) and/or motivate the choice of such statistic? How about the median (if it makes sense)?

Thank you for your suggestion. We have added the boxplots for the place and individual diversity metrics in Supplementary Figure S13. The boxplots indicate that the mean and median values are similar and robust in all cities across all periods. Thus, we will use the mean value in the result plots of the main manuscript.

6. (Result 1) It is not clear whether the monthly observations shown in Figure 1 are computed as (1) the difference of the mean of the distributions (of individuals' values) for month 2019 and month X or (2) as the mean value of the individuals' differences (month 2019 - month X). Please specify and motivate.

Thank you for clarifying. Definition (1) is correct. We have specified and motivated this in the manuscript, in Page 9 Line 296: "Most of the results in the main manuscript are shown by percentage differences, which is computed by $\Delta D_i(t) = \frac{\bar{D}_i(t) - \bar{D}_i(2019)}{\bar{D}_i(2019)} \times 100(\%)$, where $\bar{D}_i(2019)$ is the mean income diversity of encounters across all individuals observed on the same month as t in 2019, before the pandemic."

7. (Figure 1) It contains a mixture of Boston-specific results and results for other cities, and it might be confusing to the reader; I would try to separate them (maybe two separate rows?).

Thank you for your suggestion. We have explicitly added the label 'Boston' to Boston results. The results of other cities are in the supplementary material.

8. (Figure 1) I wonder whether panel A could better convey its message if plotted like the one in Figure 3, i.e., showing the difference w.r.t the baseline as well as the distribution.

Thank you for your suggestion. In Figure A, we wanted to show the spatial heterogeneity of the place diversity metrics (in baseline conditions) in addition to the dynamics across four periods. For this reason, we have kept the original figure, however, we have added plots showing the differences in Supplementary Figure S8.

9. (Figure 1) I don't quite like panel D: would it be still clear if plotted like Figure S8, i.e., with all monthly observations and not just 3? Alternatively, if authors want to show those 3 months specifically, maybe a heatmap could be fancier (x-axis: categories, y-axis: month, cell: value of the delta, with a colorbar)

Thank you for your suggestion. We wanted to keep the main figures concise and to-the-point, thus we would like to show these 3 months specifically as they convey the category-based differences effectively. We tried plotting them in a heatmap, however found that the current version aesthetically fits in better with respect to the other panels in Figure 1 (panels B and C being line plots). For completeness, we have included a line plot similar to Figure S8 in the Supplementary Material as Figure S10.

10. (Result 2): the introductory paragraph is confusing: there is an “In addition” which gives the idea that the scenarios are 4 rather than 3. Please clarify.

Thank you for pointing this out. We have edited this.

11. (Result 2): If I understand it correctly, random counterfactual datasets are simulated only once. I would expect instead N simulations and then compute the average difference with the empirical data, but I might be missing something here.

Thank you for your suggestion. We ran the counterfactual simulations 10 times each and took the mean value of the 10 runs to ensure that the differences with the empirical data are robust. The standard errors of the mean income diversity computed in the 10 simulations were extremely small compared to the actual values of income diversity due to the large number of POIs (50K~200K depending on the city) and users (140K~450K), similar to the discussion in the response to your question #3. The average of the standard error divided by the actual income diversity metrics were 0.014% and 0.017% for place and individual income diversity, respectively. We have mentioned this in Page 9 Line 311: ‘For each counterfactual scenario, the simulations were run 10 times. After creating the synthetic counterfactual datasets, we compute the income diversity of encounters and compare with the experienced income diversity measured using the actual observed data $\mathcal{D}_{y,m}$ to delineate the effects of reduction in active users and visits to place categories on the decrease in experienced income diversity. The income diversity metrics obtained from the 10 simulation runs were robust, with the average standard error of the estimated income diversity being 0.014% and 0.017% of the actual values of place and individual income diversity, respectively. The standard errors are omitted from Figure 2B since the confidence intervals are too small for visibility.’

12. (Result 2): “reduction in total activities caused around 50% of the decrease in diversity” this sentence is confusing, because the figure shows the absolute decrease of diversity, so one should see that out of -30%, -15% is accounted for by the reduction in total activities. Could the authors show the composition of the reduction – which is shown in Figure 1 – instead, alongside with the absolute values of reduction?

Thank you for your suggestion. We have added the proportion of income diversity decrease in Supplementary Figure S21.

13. (Result 2): Are results shown in Figure S14 statistically significant (“Although we observe different rates of dwell time decrease and recovery across income quantiles where the richer populations

disproportionately reduce dwell times at places than poorer populations”)? I believe it shows only the average values of POIs for each quantile, if I understand it correctly.

Thank you very much for your suggestion. We have estimated the confidence interval of the plotted values using a bootstrap approach. The new figures with bootstrapped 95% confidence intervals of mean values are shown in Supplementary Figures S17 and S19. The analysis shows that the results are statistically significant.

14. (Figure 2): I appreciate it but I don't particularly like panel A, I wonder whether it really adds much to the understanding of the results.

Thank you for your suggestion. We have revised Figure 2 given the other reviewer's comments to improve the readability of the diagram. The results in Figure 2 are the central pieces of our research, and we use panel A a number of times in the text to improve the understanding of the concept of our counterfactual analysis, which could be slightly complicated to grasp.

15. (Figure 2): How are colors assigned to categories in panel D? Please clarify.

The colors are assigned by their major POI categories, used in Figure 1D. We added “Colors correspond to the major POI categories used in Figure 1D.” in Figure 2 caption.

16. (Result 3): I am not very convinced by the approach used by the authors to find ‘predictors’ of variation in diversity, namely building a regression model for each month. I am not an expert of econometric models but I think this part needs some further motivation and/or alternative analyses. I understand that Panel regressions or ARIMA are not suitable because the socio-economic/demographic variables are fixed, but maybe carrying out a regression with all the monthly observations at once could be one direction. Also, some descriptive analyses could strengthen their findings, e.g., show that if variable X is positively associated with variation, then the top percentiles in the distribution of X should consistently exhibit a larger variation over time. Worst case, the authors could find some strong references where the same approach is employed on similar data/analyses. (Figure 3): In light of my comments above, I am a bit skeptical about panels B and C.

Thank you for your comments and questions. The objective of the analysis was to understand 1) how well we can predict (variance explained) the income diversity at places and for individuals for each month during the pandemic period, and 2) obtain a dynamic estimation of how much each factor (e.g., place individuals visit, sociodemographic characteristics of the CBG) via monthly regression coefficients. We also agree with your point and so we carried out a regression with all the monthly observations at once as you suggested. The results are mostly in agreement with the original regression modeling setup, where for D_{cbg} , the residential demographics variables account for the majority (65%) of the explained variance, and for ΔD_{cbg} , both residential demographics and places visited account for 40% each. The results for the pooled regression are shown in the Supplementary Material, in Figures S28 - S33 and in Tables S11, S12.

17. (Result 4): Authors say “we tested ARIMA type models as well, however the regression results and especially the estimated coefficients were found to be robust” but a double check with Table S14 does not show this (cf. Los Angeles and Dallas). Please clarify.

Thank you for pointing this out. We have revised the wording of this in the manuscript. Page 8 Line 218 reads: “Since $\Delta D_{\text{CBSA}}(t)$ is a temporal data with autocorrelation, we tested ARIMA type models as well. For Boston and Seattle, the moving average component was significant, however the estimated coefficients of the stringency index were found to be robust. The temporal effects for Los Angeles and Dallas were insignificant (see Supplementary Note 6.2, Supplementary Tables S13 and S14, and Supplementary Figure S25).”

18. (Conclusions): I think the authors expand a bit more on the limitations of their study.

Thank you for your suggestion. We have expanded on the limitations of our study, especially on the limitations of the data with additional references. Page 8 Line 253 reads: “Although we have tested the robustness of the used mobile phone data from various aspects and applied a post-stratification strategy to overcome sampling biases, there may be overlooked biases arising from uncertainty in data collection algorithms, such as the frequency and timing of location data collection [54,55].”

Minor:

19. Can the authors motivate the choice of the four cities (although I believe it’s more a matter of data availability)?

The limited number of cities focused in the study is mainly due to limitations in data access and computational resources. Strict limitation to data access is imposed by our data providers, to ensure ethical and legal use of the data in a privacy-preserving platform. Given these constraints, the four cities were selected to maximize the diversity of characteristics in terms of sociodemographic details, political inclinations, weather characteristics, COVID-19 policy strictness, etc.

20. I recommend adding a few pointers to the literature related to the (broad) socio-economic effects of the pandemic (including NPIs) inferred via mobility data as well as the epidemiological literature on the effects of mobility restrictions. I will not provide specific references to avoid biases, and I am sure the authors can find appropriate citations

Thank you for your suggestion. Together with similar comments from reviewer 2, we have added key literature related to the socioeconomic effects of the pandemic inferred using mobility data, and how they were used for epidemiological modeling.

Page 2, Line 36 now reads: “There has been many studies in the fields of civil engineering, geography, and urban planning that have analyzed the short-term mobility dynamics during the early stages of the pandemic and lockdown [30-33], socioeconomic disparities in response to lockdowns [34-36], and its impact on disease spread using location data collected via mobile phones [37,38].”

21. I recommend ensuring more consistency between the manuscript and the supplementary material, i.e., please make sure to reference figures and tables in the order they are mentioned in the main text, as much as possible.

Thank you for your suggestion. We agree that we lacked consistency between the manuscript and the Supplementary Material. On the other hand, we do want to keep a coherent flow of logic in the SI. For the

sake of clarity, we have decided to limit the number of citations we make to specific Supplementary Figures and Tables from the main manuscript, and to point to only the section numbers in the Supplementary Notes.

22. I recommend making another round of proof-reading: there are a few typos starting from the abstract (e.g. inconsistencies of verbs with singular/plural/third person) and some sentences sound a bit wordy (e.g. "this agrees with ..")

Thank you for pointing this out. We have fixed the typos, and revised wordy sentences (e.g., 'this agrees with' → 'this is consistent with').

23. The GitHub repository cannot be accessed (and figures cannot be reproduced) , but I doubt this happened on purpose. Please put it back online.

Apologies for this, we mistakenly made the repository private. We have put the repository back online. You can access it at <https://github.com/takayabe0505/IncomeDiversity>

REVIEWER COMMENTS

Reviewer #1 (Remarks to the Author):

The authors have addressed my comments and I do not have additional questions or comments. I recommend the publication of the manuscript.

Ryan W

Reviewer #2 (Remarks to the Author):

I greatly appreciate the authors' efforts in addressing my comments and other reviewers' comments. However, unfortunately, I do have several remaining comments that have not been resolved fully.

[Reviewer 2, Comment 1] Thank you for your response. I do hope to emphasize that there are several existing studies that investigated the longitudinal aspect of mobility changes during the pandemic. I recommend the authors briefly introduce these works (that had the longitudinal approach and did not focus only on the early phase of the pandemic). Then, the authors can emphasize the significant gaps in these existing studies and highlight their study's originality and added value to the literature.

Concas, S., Kourtellis, A., Kummetha, V., Kamrani, M., Rabbani, M., & Dokur, O. (2022). Longitudinal Study of the COVID-19 Pandemic Impact on Activity Travel Using Connected Vehicle Data. *Transportation Research Record*, 03611981221107006.

Matson, G., McElroy, S., Lee, Y., & Circella, G. (2022). Longitudinal Analysis of COVID-19 Impacts on Mobility: An Early Snapshot of the Emerging Changes in Travel Behavior. *Transportation Research Record*, 03611981221090241.

Molloy, J., Schatzmann, T., Schoeman, B., Tchervenkov, C., Hintermann, B., & Axhausen, K. W. (2021). Observed impacts of the Covid-19 first wave on travel behaviour in Switzerland based on a large GPS panel. *Transport Policy*, 104, 43-51.

[Reviewer 2, Comment 2] Thank you for your response. The authors responded that "Given these constraints, the four cities were selected to maximize the diversity of characteristics in terms of geographical locations, sociodemographic details, political inclinations, weather characteristics, COVID-19 policy strictness, etc." Although I understand that data availability is one of the important factors for selecting these four cities, the authors' response does not provide strong rationales for studying these four cities. For example, in which way do these four cities "maximize the diversity of characteristics?" If this is the case, the authors may consider providing specific examples and pieces of evidence (e.g., statistics from the Census survey data).

[Reviewer 2, Comment 3] Thank you for your response. Although I acknowledge the great utility of GPS data for COVID-19 research, including epidemiological modeling, my point here is how and to what extent the authors are confident with a specific mobile phone-based mobility dataset that is used for this study. In other words, how did the authors check the validity of their dataset regarding the accuracy in detecting granular location change in high-density urban environments? Moreover, the authors have not addressed my comment related to an example of people visiting the high-density mixed land-use area (e.g., commercial districts, such as the Prudential Center in Boston). I argue that this methodological issue is very important and deserves attention because the authors' analysis includes a detailed classification of trip purposes (e.g., grocery store, hardware, big box store, and so on, in Figure 2). To the best of my knowledge, mobile phone-based mobility data is highly limited to capturing detailed classification of trip purposes especially when study areas include high-density mixed land-use areas. It is still unclear how the authors have addressed this important methodological issue.

[Reviewer 2, Comment 5] Thank you for addressing this comment. Figure 1A becomes better as it provides background geographic information on the map. However, Figure 3A can be improved more by adding some labels (of landmarks) because the map in its current form is less interpretative in terms of geographic information.

Reviewer #3 (Remarks to the Author):

I am not completely satisfied with how the authors addressed some of my concerns, and I think the authors can still improve the quality of the manuscript before it gets published. In particular, I'd like the authors to take another shot at my previous comments: (1), (2), (9), and (22). These all require minor modifications which I believe are needed for a publication in Nature Communications.

Responses to the Reviewers for “Behavioral changes during the pandemic worsened income diversity of urban encounters”

Takahiro Yabe, Bernardo Garcia Bulle Bueno, Xiaowen Dong, Alex ‘Sandy’ Pentland, and Esteban Moro

We thank the reviewers for their helpful comments, questions, and suggestions and for their thorough review of our manuscript. We have extensively revised the manuscript based on the provided reviews to address their concerns. We list the revisions made and responses to each comment provided by the reviewers below.

Reviewer #1:

The authors have addressed my comments and I do not have additional questions or comments. I recommend the publication of the manuscript.

We greatly appreciate your constructive and positive comments and suggestions.

Reviewer #2:

I greatly appreciate the authors' efforts in addressing my comments and other reviewers' comments. However, unfortunately, I do have several remaining comments that have not been resolved fully.

[Reviewer 2, Comment 1] Thank you for your response. I do hope to emphasize that there are several existing studies that investigated the longitudinal aspect of mobility changes during the pandemic. I recommend the authors briefly introduce these works (that had the longitudinal approach and did not focus only on the early phase of the pandemic). Then, the authors can emphasize the significant gaps in these existing studies and highlight their study's originality and added value to the literature.

- Concas, S., Kourtellis, A., Kummetha, V., Kamrani, M., Rabbani, M., & Dokur, O. (2022). Longitudinal Study of the COVID-19 Pandemic Impact on Activity Travel Using Connected Vehicle Data. *Transportation Research Record*, 03611981221107006.
- Matson, G., McElroy, S., Lee, Y., & Circella, G. (2022). Longitudinal Analysis of COVID-19 Impacts on Mobility: An Early Snapshot of the Emerging Changes in Travel Behavior. *Transportation Research Record*, 03611981221090241.
- Molloy, J., Schatzmann, T., Schoeman, B., Tchervenkov, C., Hintermann, B., & Axhausen, K. W. (2021). Observed impacts of the Covid-19 first wave on travel behaviour in Switzerland based on a large GPS panel. *Transport Policy*, 104, 43-51.

Thank you for your suggestion. We agree that Concas et al. (2022) in particular investigates the longitudinal aspects of mobility changes during the pandemic until May 2021. Matson et al. (2022) and Molloy et al.

(2021) are additional works that investigate the short term (first wave of the pandemic) mobility dynamics during the pandemic, until early 2020 and August 2020, respectively, which we have added to our introduction. We would like to point out that the critical gap in the literature that we aim to bridge in this work is the lack of research in how such mobility and behavioral changes during the pandemic has affected the quantity and quality of our physical encounters in cities. We have emphasized this point in the Introduction.

[Reviewer 2, Comment 2] Thank you for your response. The authors responded that "Given these constraints, the four cities were selected to maximize the diversity of characteristics in terms of geographical locations, sociodemographic details, political inclinations, weather characteristics, COVID-19 policy strictness, etc." Although I understand that data availability is one of the important factors for selecting these four cities, the authors' response does not provide strong rationales for studying these four cities. For example, in which way do these four cities "maximize the diversity of characteristics?" If this is the case, the authors may consider providing specific examples and pieces of evidence (e.g., statistics from the Census survey data).

Thank you for your suggestion. To quantitatively show the diversity of the four cities' characteristics, we plot various sociodemographic, geographic, COVID-19 related, and climate related variables of the four cities among the largest 15 metropolitan statistical areas in the US. More specifically, in Figure S6 we show the population (log), population % change from 2020 to 2021, median household income, poverty rates, % of workers who use private vehicles to work, % population with education level with Bachelor degree (all using the American Community Survey), the COVID-19 stringency index in 2022 January provided by the University of Oxford, and the annual average temperature of the four cities compared against the largest 15 metropolitan statistical areas in the US. The figure shows that our collection of four cities covers a wide range of values for each of the sociodemographic, political (COVID-19 stringency), and climate characteristics.

[Reviewer 2, Comment 3] Thank you for your response. Although I acknowledge the great utility of GPS data for COVID-19 research, including epidemiological modeling, my point here is how and to what extent the authors are confident with a specific mobile phone-based mobility dataset that is used for this study. In other words, how did the authors check the validity of their dataset regarding the accuracy in detecting granular location change in high-density urban environments? Moreover, the authors have not addressed my comment related to an example of people visiting the high-density mixed land-use area (e.g., commercial districts, such as the Prudential Center in Boston). I argue that this methodological issue is very important and deserves attention because the authors' analysis includes a detailed classification of trip purposes (e.g., grocery store, hardware, big box store, and so on, in Figure 2). To the best of my knowledge, mobile phone-based mobility data is highly limited to capturing detailed classification of trip purposes especially when study areas include high-density mixed land-use areas. It is still unclear how the authors have addressed this important methodological issue.

We thank the reviewer for highlighting this important point. During the last years, we have extensively tested the suitability of the specific high-precision mobility dataset (note that the dataset contains GPS signals with spatial noise of ~10 meters, as opposed to cell tower records which have substantial larger spatial noise especially in rural areas) used in this study to investigate different urban behaviors. We have used it in different contexts and lately for COVID-19 research. We have validated different aspects of it, from its population representativeness, visit attribution, and even travel behavior. For example in our previous work (Moro et al; 2021) and subsequent works (Hunter et al; 2021, Aleta et al; 2020, Aleta et al; 2022) we have shown that the dataset is highly representative of the different socio-demographic groups in the urban areas considered. Even more interestingly, our results about changes in visitation patterns to different places during the pandemic (e.g. groceries or food places, see Aleta et al; 2020 and Aleta et al; 2022) and subsequent infections coincide with other works based on different datasets, surveys, and epidemiological data (see those papers for references). Even the results about the decrease in walking behavior detected from this dataset coincide with Apple or Google mobility trends (see Hunter et al; 2021). Since Spectus provided access to this dataset through their Data for Good initiative we are not the only group using this dataset and other research groups, agencies and governments have used it and validated it in many other different contexts (see e.g., World Bank blog on use of Spectus data for pandemic response).

We admit there may be spatial noise that could affect the visit attribution process in high-density mixed land-use areas such as the Prudential Center in Boston, even with the ~10m granularity accuracy of mobility data. Unfortunately, due to limitations in ground truth data, it is infeasible to validate all the actual visit counts to individual stores. However, a previous study (Moro et al; 2021) checked the representativeness of the same dataset at the level of major sports venues, including games in the National Football League (NFL), National Basketball Association (NBA), and National Hockey League (NHL), and showed high accuracy in estimating the attendance for each game, validated using official statistics. While this validation was conducted on large-scale venues, accurate estimations for NBA and NHL venues in urban locations (which are high-density and mixed-use), such as TD Garden in Boston or the Madison Square Garden in NYC, provide strong evidence that the mobile phone location data may be used to accurately estimate visitation patterns to places (Supplementary Material for Moro et al., 2021).

To address further the comments by the reviewer we have investigated the effect of our visit attribution in high-density areas in our results. We find that such high density POIs, which have 5 or more POIs located within 20 meters including vertical alignment (which is a generous spatial threshold relative to the GPS location error of ~10 meters), account for around 5% of POIs and 2% of total dwell time spent, averaged across the four cities. Figure S8 shows the percentage of high density POIs and the proportion of total dwell time spent in those locations, under different parameters for determining high density POIs (3, 5, 10, or 20 POIs located within 20 meters). This shows that although the visit estimation to high density POIs could be less accurate, the absolute number of high density POIs and the total number of visits to those POIs are relatively small.

To investigate whether this issue does not affect the overall results on the income diversity of urban encounters, we recomputed the results using two methods: 1) compute the diversity at places without the inclusion of such high density POIs, and 2) refine our visit attribution algorithm for such high density POIs using the Foursquare checkin data as a proxy for the likelihood of visiting different POIs. More specifically, for visits to high density POIs (those with 3 or more POIs located within 20 meters) visits are attributed to POIs proportionally to the number of check-ins each POI has on the Foursquare platform (see details in

Supplementary Note 2.3). This method allows us to refine and improve the visit attribution using a different dataset generated by spatially granular human activity patterns. In both methods and in all four cities, we found the dwell time spent at different categories are consistent with the original data (Pearson $R > 0.99$) and the estimated income diversity values of each place category (e.g., grocery store, hardware, big box store) are also consistent with the original estimation (Pearson $R > 0.975$). From these results under different threshold parameters and validation approaches, we conclude that the results on income diversity are robust against the treatment of high density POIs. Detailed results are shown in Supplementary Note 2.3.

[Reviewer 2, Comment 5] Thank you for addressing this comment. Figure 1A becomes better as it provides background geographic information on the map. However, Figure 3A can be improved more by adding some labels (of landmarks) because the map in its current form is less interpretative in terms of geographic information.

Thank you for your suggestion, we have added more labels of landmarks in Figure 3A.

Reviewer #3:

I am not completely satisfied with how the authors addressed some of my concerns, and I think the authors can still improve the quality of the manuscript before it gets published. In particular, I'd like the authors to take another shot at my previous comments: (1), (2), (9), and (22). These all require minor modifications which I believe are needed for a publication in Nature Communications.

1. Similar to several papers published in NC (as far as I've seen), I would add a few sentences on motivation/gaps in the literature/results/conclusions at the end of the introduction, to provide a general idea of the paper to the reader

Thank you for your suggestion. We have added a short paragraph at the end of the introduction to provide a general idea of the paper to the reader.

The final paragraph of the introduction now reads: “To this end, this study contributes to bridging this knowledge gap by analyzing how the income diversity of urban encounters have changed before and during the pandemic across a 3 year timespan, using a large-scale, privacy-enhanced mobility dataset of more than one million anonymized mobile phone users in four large US cities. We find that the diversity of urban encounters has substantially decreased during the pandemic and has persisted through late 2021, despite the recovery of aggregated mobility metrics. Furthermore, we conduct counterfactual analyses to show that behavioral changes including lower willingness to explore new places and changes in visitation preferences substantially worsened the long-term diversity of encounters. Our findings suggest that the pandemic could have long-lasting negative effects on experienced income diversity, and call for substantial efforts and policies to recover the diversity of urban encounters as we move beyond the pandemic.”

2. I recommend the author to review captions of figures: they should first describe the content shown and then make conclusions, without requiring the reader to go back and forth from the text to the figures. Also, I would try to be more precise in the Supplementary: many captions do not clearly tell what they show.

Thank you for your suggestions. We have substantially reviewed the captions of the figures both in the main manuscript and the supplementary material. In particular, we have substantially revised the captions of the figures in the Supplementary material to improve the clarity of the figures (See Figures S1 - S18, S20, S23 - 24, S30, S38).

9. (Figure 1) I don't quite like panel D: would it be still clear if plotted like Figure S8, i.e. with all monthly observations and not just 3? Alternatively, if authors want to show those 3 months specifically, maybe a heatmap could be fancier (x-axis: categories, y-axis: month, cell: value of the delta, with a colorbar)

Thank you for your suggestion. We changed Figure 1D to a line plot similar to Figure S8 which includes the monthly observations, not just 3. We have similarly changed the format of the plots for the other cities in the Supplementary Material (Figures S13).

22. I recommend making another round of proof-reading: there are a few typos starting from the abstract (e.g. inconsistencies of verbs with singular/plural/third person) and some sentences sound a bit wordy (e.g. "this agrees with ..")

Thank you for your suggestion, we have gone through another round of proofreading.

REVIEWERS' COMMENTS

Reviewer #2 (Remarks to the Author):

The authors have addressed all my comments. I greatly appreciate the responsiveness of the authors to my prior comments.

Responses to the Reviewers for “Behavioral changes during the pandemic worsened income diversity of urban encounters”

Takahiro Yabe, Bernardo Garcia Bulle Bueno, Xiaowen Dong, Alex Pentland, and Esteban Moro

We thank all the reviewers for their helpful comments, questions, and suggestions and for their thorough review of our manuscript.

Reviewer #2:

The authors have addressed all my comments. I greatly appreciate the responsiveness of the authors to my prior comments.

We thank the reviewer for the helpful comments and suggestions.